# Visualizing orthogonal RNAs simultaneously in live mammalian cells by fluorescence lifetime imaging microscopy (FLIM)

Nadia Sarfraz [1], Emilia Moscoso [1], Therese Oertel[1], Harrison J. Lee [1], Suman Ranjit [2,3] & Esther Braselmann [1] ✉

Visualization of RNAs in live cells is critical to understand biology of RNA dynamics and function in the complex cellular environment. Detection of RNAs with a fluorescent marker frequently involves genetically fusing an RNA aptamer tag to the RNA of interest, which binds to small molecules that are added to live cells and have fluorescent properties. Engineering efforts aim to improve performance and add versatile features. Current efforts focus on adding multiplexing capabilities to tag and visualize multiple RNAs simultaneously in the same cell. Here, we present the fluorescence lifetime-based platform Riboglow-FLIM. Our system requires a smaller tag and has superior cell contrast when compared with intensity-based detection. Because our RNA tags are derived from a large bacterial riboswitch sequence family, the riboswitch variants add versatility for using multiple tags simultaneously. Indeed, we demonstrate visualization of two RNAs simultaneously with orthogonal lifetime-based tags.

Quantifying RNA localization and subcellular dynamics within living cells provides critical insights in RNA function, motivating the need for robust RNA fluorescence labels[1]. Genetically encoded RNA tags that bind a small molecule and induce light-up fluorescence have gained popularity for use in mammalian cells (Supplementary Table 1). Features have been systematically optimized and added, including advantageous RNA-probe interactions[2,3] and color-shifting properties for ratiometric fluorescence readout[4]. Ratiometric sensors are intensity-independent and avoid concentration-dependent artifacts. Spectrally distinct RNA sensors allow labeling multiple RNAs of interest simultaneously[5], but cross-reactivity between probes and RNA tags is a concern, as observed for sensors built from the Broccoli family[6-8]. To capture the complexity of RNA processes in live cells, engineering tags to dissect subcellular RNA dynamics and visualize multiple RNAs simultaneously is critical.

Fluorescence lifetime yields quantitative, concentration-independent readouts of reporters for fluorescence lifetime imaging microscopy (FLIM) to track dynamics of labeled species live[9]. Sensors can be engineered to fluoresce in the same spectral range but exhibit different lifetimes for simultaneous multiplexed sensing in live cells[10]. Exploiting fluorescence lifetime for intensity-independent sensing of multiple different RNAs simultaneously is the foundation of this study, where we expand the genetically encoded Riboglow RNA sensor platform[11]. First, we establish Riboglow-FLIM as a superior approach for live RNA visualization compared with intensity-dependent sensing. We find that Riboglow-FLIM yields higher contrast with reduced RNA tag size. Second, we demonstrate robust quantification of subcellular RNA species live with our Riboglow-FLIM platform. Finally, we chose two different Riboglow tag variants and assessed orthogonality of imaging two different RNAs simultaneously. We demonstrate that two RNAs with distinct localizations in live mammalian cells may be visualized simultaneously with our Riboglow-FLIM approach. This goal was not previously achieved with aptamer-based RNA fluorescent tags.

Riboglow is a genetically encoded RNA tagging platform consisting of a riboswitch-derived RNA tag and a small probe, Cobalamin (Cbl) coupled to a synthetic fluorophore (Fig. 1, Supplementary

[1]Department of Chemistry, Georgetown University, Washington, DC, USA. [2]Department of Biochemistry and Molecular & Cellular Biology, Georgetown University, Washington, DC, USA. [3]Microscopy & Imaging Shared Resource, Georgetown University, Washington, DC, USA. ✉ e-mail: esther.braselmann@georgetown.edu

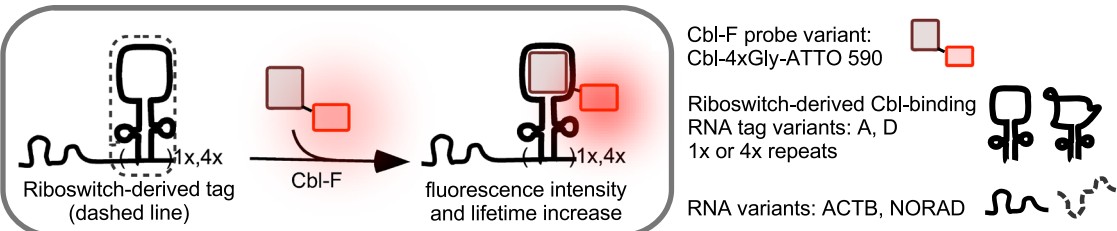

**Fig. 1 | Visualizing RNAs in live mammalian cells using the modular Riboglow platform.** Cobalamin (Cbl) is synthetically coupled to a fluorophore (F) via an organic linker. Fluorescence of F is quenched by Cbl in the Cbl-F context. The Cbl moiety binds to a specific riboswitch-derived RNA sequence, inducing fluorescence intensity and fluorescence lifetime increase in Cbl-F. For live cell fluorescence detection, the RNA tag is genetically fused to an RNA of interest (e.g., ACTB mRNA, NORAD). Riboglow RNA tags with affinity to Cbl may vary in sequence (here, variants A and D). One or multiple RNA tag repeats may be used.

Table 2)[11]. The RNA tag is derived from the Cbl-binding riboswitch family[12] and the fluorophore is variable without interfering with RNA-binding[11]. Cbl is a fluorescence quencher; hence, the Cbl-fluorophore probe is dim in the RNA-unbound state, as confirmed previously when Riboglow was developed as an intensity-based RNA imaging system[11]. Fluorescence intensity increases upon RNA binding for live cell fluorescence microscopy[11]. Here, we hypothesized that a concomitant increase in probe fluorescence lifetime upon RNA binding may establish Riboglow for FLIM-based, intensity-independent RNA detection with multiplexing capabilities.

## Results

### Validation of Riboglow's fluorescence lifetime in vitro
Our hypothesis that visualizing Riboglow by FLIM (Riboglow-FLIM) is advantageous to achieve robust contrast for live RNA sensing is based on the prior observation that free Cbl-5xPEG-ATTO 590 exhibited a substantial increase in fluorescence lifetime in vitro when bound to purified Riboglow RNA[11]. To establish Riboglow-FLIM as a live cell imaging application, we first asked if a similar change in fluorescence lifetime was detected in our hands. We indeed observed a strong increase in fluorescence lifetime for Cbl-4xGly-ATTO 590 in the presence of purified Riboglow RNA A and D (Supplementary Fig. 1). Notably, lifetime values for the probe when bound to the A tag vs. the D tag differed, pointing to the idea to differentiate RNA tags via their lifetime (Supplementary Fig. 1c). As expected, the probe moiety Cbl that facilitates RNA / probe binding was required to change fluorescence lifetime of ATTO 590 (Supplementary Fig. 2). Together, our in vitro measurements lead us to predict that live mammalian cells that produce the Riboglow RNA sequence will alter the fluorescence lifetime of the Riboglow probe and enable RNA detection by fluorescence lifetime imaging microscopy (FLIM).

### Riboglow-tagged mRNA localization in cell model
We chose mRNA that encodes for β-actin (ACTB) as a model mRNA for Riboglow-FLIM in live U-2 OS cells, as it was previously established to evaluate Riboglow performance[11]. We confirmed that localization of ACTB mRNA tagged with four copies of the Riboglow sequence and produced from a plasmid is indistinguishable from endogenous ACTB mRNA (Supplementary Fig. 3). Localization patterns of our model RNA reporters are indistinguishable when probed against the RNA vs. Riboglow tag sequence, confirming that the Riboglow tag does not induce mislocalization artifacts (Supplementary Fig. 3). Together, we concluded that our live cell detection will be sensitive to visualize ACTB mRNA tagged with Riboglow by FLIM.

### FLIM data processing by multiexponential reconvolution
We established a FLIM workflow for Riboglow-FLIM visualization in live mammalian cells as follows. First, we robustly reproduced fluorescence lifetime values of a thoroughly characterized fluorescent marker, the mCherry protein (Supplementary Fig. 4)[13–16]. We then performed FLIM for live U-2 OS cells that were transfected with a plasmid to produce the Riboglow-tagged reporter mRNA ACTB-Ribo(4D)−590 and loaded with the Riboglow probe, a model that was established previously[11]. We included a transfection marker to identify cells that produced Riboglow-tagged RNA. For FLIM data processing, each whole cell was defined as a region of interest (ROI). The unfit lifetime data (FastFlim image) for each ROI was processed by multi-exponential reconvolution (Fig. 2a, Supplementary Fig. 5). Each ROI can then be further processed in two different ways. First, the average fluorescence intensity and amplitude-weighted lifetime from the fluorescence decay (Supplementary Fig. 6, Supplementary Note 1) yields one lifetime number for each ROI. We call this number the "average lifetime" (Fig. 2) that we use to compare lifetimes between cell populations, represented on a dot plot as "Lifetime" (Fig. 2a, and figures throughout this manuscript). Alternatively, the "component lifetime" resolves fluorescence lifetimes pixel-by-pixel for visual representation of microscopy images (Fig. 2a). In both cases, we use a false-color scale to illustrate lifetime values (Fig. 2b). Both analyses yielded a substantial increase in fluorescence lifetime for cells that produced the Riboglow-tagged reporter mRNA vs. an untransfected control (Fig. 2c).

### Assessing Riboglow-FLIM in cytosol and nucleus
To evaluate performance of Riboglow-FLIM, we compared cells that produced ACTB-Ribo(4D)−590 vs. control cells that were not transfected with the reporter, and only the Riboglow probe was loaded into cells (Fig. 2). Importantly, the contrast between cells that produce the Riboglow reporter and untransfected control cells is much greater for FLIM visualization vs. intensity-based imaging (Fig. 3). Together, we conclude that FLIM is a robust imaging modality for visualizing RNAs tagged with the Riboglow platform.

We systematically assessed whether the FLIM signal distribution for Riboglow-tagged mRNA (ACTB-Ribo(4D)−590) across the whole cell accurately reports on subcellular mRNA localization. Bead loading the probe into live cells did not affect ACTB mRNA localization (Supplementary Fig. 7). Fluorescence lifetime of the probe alone was slightly elevated in the nucleus, as observed in pixel-by-pixel representation (Fig. 2). This behavior is in contrast to intensity-based imaging, where substantial non-specific nuclear probe localization was observed, likely due to probe concentration-dependent artifacts. A similar concentration-dependent nuclear Riboglow probe signal was observed previously as well[11]. We quantified this observation systematically and observed a detectable but not significant increase in nuclear fluorescence lifetime (Supplementary Fig. 8). Notably, this effect was much smaller than robust differences in lifetime for the experimental vs. control cells, and much more pronounced for fluorescence intensity quantification (Supplementary Fig. 8). While the cause of non-specific nuclear signals remains unclear, we are encouraged that FLIM substantially and quantitatively reduced this artifact.

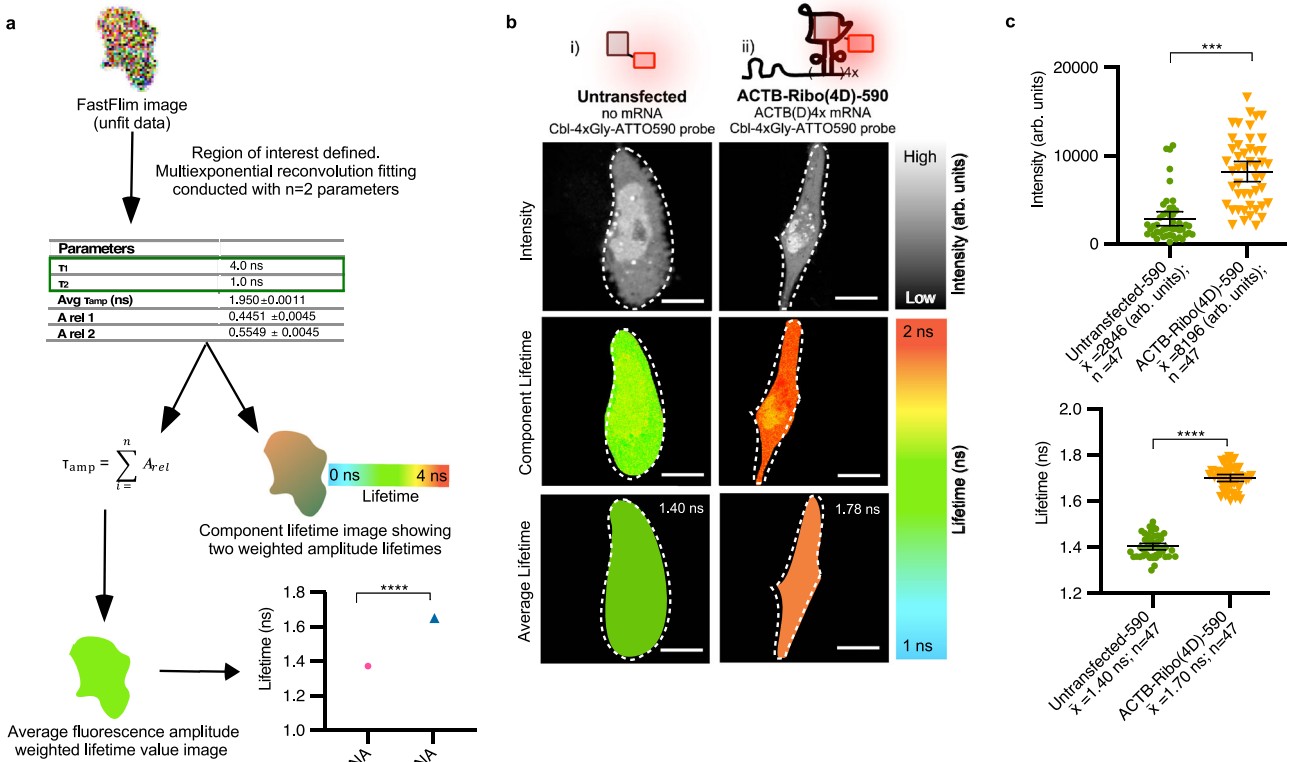

**Fig. 2 | Principle of FLIM data processing. a** Workflow to analyze fluorescence lifetime images by multiexponential reconvolution fitting. Live mammalian cells are transfected with the Riboglow reporter, a transfection marker, and loaded with the Riboglow probe. Time-Correlated Single Photon Counting (TCSPC) is used to acquire a FastFlim image using SymPhoTime 64 (Picoquant) for a specific region of interest (ROI) in live mammalian cells, resulting in the unfit raw lifetimes of each individual pixel. This data then undergoes a multiexponential reconvolution fitting, producing representative quantitative data such as lifetime values ($\tau_1$ and $\tau_2$) for further processing. The average fluorescence weighted lifetime value represents an average value for the entire ROI where each ROI is defined as an entire cell. This analysis is used for dot plots throughout this study. Alternatively, the component lifetime image yields the lifetime value on a pixel-by-pixel basis, useful for image representations. **b** Example of different lifetime representations for two cells with different variants of the Riboglow reporter. Lifetime values can be represented using a false-color scale for the component lifetime or average lifetime representation vs. intensity-based detection. A ROI was defined as the entire cell (estimated by a dotted line). Fluorescence intensity was quantified pixel-by-pixel and the background was subtracted. Lifetime of the representative cell is indicated. Scale bar = 10 μm. **c** A whole-cell was defined as an ROI and fluorescence intensity or fluorescence lifetime were extracted (6 independent experiments, 94 cells, 1 symbol = 1 cell). One-way ANOVA (95% confidence limit); post hoc test (Tukey HSD), ***$p \le 0.001$, ****$p \le 0.0001$. Error bars indicate mean and standard deviation (+/−SD).

## Comparing FLIM data analysis approaches confirms robustness

Having confirmed that Riboglow-FLIM accurately reports on subcellular localization of a reporter mRNA, we compared different FLIM data analysis approaches. We observed a robust increase in fluorescence lifetime for both component average lifetime (Fig. 2a) and average lifetime (Fig. 2a, dot plot) for Riboglow-tagged RNA vs. probe alone. The fit-free phasor approach (Supplementary Fig. 9, Supplementary Note 2) and tailfit analysis (Supplementary Table 3) similarly revealed robust differences between untransfected control vs. Riboglow-producing cells. Together, Riboglow-FLIM allows for robust RNA sensing in live cells independent of how the data is analyzed.

## Riboglow-FLIM allows for minimal tag size and orthogonality

Next, we systematically and quantitatively assessed Riboglow-FLIM capabilities. We evaluated the potential for visualizing two different RNAs simultaneously. For this, we compared cells that produced RNAs labeled with two distinct members of the Cbl-riboswitch family: ACTB-Ribo(4 A)−590 vs. ACTB-Ribo(4D)−590 mRNA (Fig. 3). We observed statistically significant differences in fluorescence lifetime for both tags, in line with in vitro lifetime measurements (Supplementary Fig. 1). These differences were not discernable using fluorescence intensity (Fig. 3). To evaluate the robustness of this observation, we used a blind analysis where cells were loaded with the probe and produced

Riboglow-tagged mRNA (variant A or D), or remained untransfected. We found that the cellular lifetime values alone are sufficient to unambiguously assign which RNA tag was present, or if cells were untransfected (Supplementary Fig. 10). For further expansion of Riboglow-FLIM capabilities, we then reduced the tag to one RNA tag copy, yielding ACTB-Ribo(1A)−590 (Fig. 3). Strikingly, we observed high fluorescence lifetime contrast vs. the untransfected control, while intensity differences were not detectable with only a single RNA tag copy (Fig. 3). Together, Riboglow-FLIM enables usage of a minimally perturbing RNA tag (-100 nt for 1xA tag), and presents the possibility of multiplexed RNA imaging.

## Riboglow-FLIM readout is cell type- and reporter-independent

We compared performance of Riboglow-FLIM across different cell types and when tagged to a series of different reporter RNAs. First, ACTB-Ribo(4A)−590 was evaluated in comparison vs. untransfected control cells for U-2 OS cells, HeLa cells, and the breast cancer model cell line MDA-MB231 (Fig. 4a). We did not find cell line-specific differences in fluorescence lifetime. Next, we compared tagging different non-coding RNAs vs. ACTB mRNA with the same Riboglow tag, namely four copies of the Riboglow A tag (Fig. 4b). No changes in fluorescence lifetime were observed when different RNAs were tagged with Riboglow. Together, we concluded that Riboglow-FLIM is a versatile tool

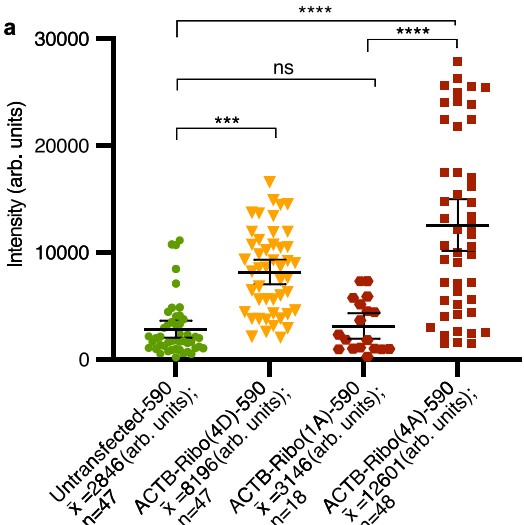

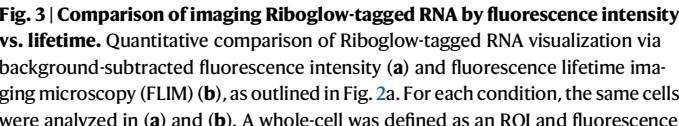

**Fig. 3 | Comparison of imaging Riboglow-tagged RNA by fluorescence intensity vs. lifetime.** Quantitative comparison of Riboglow-tagged RNA visualization via background-subtracted fluorescence intensity (**a**) and fluorescence lifetime imaging microscopy (FLIM) (**b**), as outlined in Fig. 2a. For each condition, the same cells were analyzed in (**a**) and (**b**). A whole-cell was defined as an ROI and fluorescence intensity (**a**) or average fluorescence lifetime (as defined in Fig. 2) (**b**) were extracted (6 independent experiments, 160 cells, 1 symbol = 1 cell). (ns: $p \leq 0.5$; *$p \leq 0.05$; **$p \leq 0.01$; ***$p \leq 0.001$; ****$p \leq 0.0001$, see Supplementary Table 4). One-way ANOVA (95% confidence limit); post hoc test (Tukey HSD). Error bars indicate mean and standard deviation (+/−SD).

for live RNA visualization with robust performance across cell types, independent of the RNA that is tagged.

### Stress granules as a model for subcellular RNA localizations

As a model application, we used RNA recruitment to stress granules (SGs) to assess subcellular RNA localization. SGs form in response to stress and contain RNAs and proteins, including the SG marker G3BP1[17,18]. We used U-2 OS cells that produce Halo-tagged G3BP1 from the chromosome to identify SGs (Supplementary Fig. 11, Supplementary Fig. 12). We indeed observed robust SG formation upon arsenite treatment (Supplementary Fig. 13). We confirmed that ACTB mRNA localizes to SGs by fluorescence in situ hybridization (FISH), both for endogenous ACTB mRNA and Riboglow-tagged ACTB mRNA (Supplementary Fig. 13). We then assessed if ACTB-Ribo(4D)-590 mRNA localizes to arsenite-induced SGs live. We used the cellular lifetimes of untransfected cells and cells producing Riboglow-tagged mRNA as references (Fig. 5, colored lines). By this metric, ACTB-Ribo(4D)-590 localized to SGs (Fig. 5, Supplementary Fig. 14) and a substantial portion remained cytosolic, as observed for endogenous ACTB mRNA in fixed cells[19]. We found that quantifying mRNA recruitment to SGs is more efficient by FLIM vs. fluorescence intensity (Supplementary Fig. 14).

To expand Riboglow's capability for quantitatively visualizing RNAs with distinct subcellular localizations, we tagged truncations of the long noncoding RNA NORAD with Riboglow. NORAD exhibits cytosolic localization[20], as we confirmed by FISH in our hands (Supplementary Fig. 3). Previous work showed that NORAD partitions to SGs with different efficiency depending on truncation status, with notable differences for 1/8-NORAD (exclusion from SGs) and 1/2-NORAD (recruitment to SGs)[20]. We confirmed that Riboglow-tagged 1/8-NORAD, produced from a plasmid, localizes to the cytosol (Supplementary Fig. 3) and is excluded from SGs when compared with full-length endogenous NORAD by FISH in U-2 OS cells (Supplementary Fig. 13). Together, we concluded that the NORAD model system is ideal to assess subcellular RNA localizations with Riboglow-FLIM.

### Riboglow-FLIM is a concentration-independent modality

As for ACTB-Ribo(4A)-590, Riboglow-tagged 1/2-NORAD and 1/8-NORAD was readily detected throughout the cytosol via fluorescence lifetime in unstressed cells (Fig. 3b). Increasing the RNA concentration

by transfecting more plasmid DNA encoding for the RNA reporter led to an increase in fluorescence intensity values that were highly heterogeneous between cells. In contrast, the cellular lifetime did not increase substantially (Supplementary Fig. 15). Thus, we concluded that FLIM is a robust approach for quantitatively assessing subcellular RNA localizations when Riboglow-tagged RNAs are produced from a transfected plasmid, as in our NORAD model system.

### NORAD truncation illustrates distinct localization patterns

We next evaluated quantification of NORAD truncation recruitment to SGs live. The lifetime values for 1/2-NORAD-Ribo(4A)-590 for ROIs corresponding to SGs and cytosol (Fig. 6) confirm that 1/2-NORAD-Ribo(4A)-590 localizes to SGs and a substantial portion remains cytosolic, as observed before[20]. This effect was independent of the fluorescent marker of G3BP1, excluding the possibility of fluorescence bleed-through artifacts (Supplementary Fig. 16). The ROI analysis for cells producing 1/8-NORAD-Ribo(4A)-590 yields SG lifetime values similar to untransfected cells (i.e., no Riboglow-tag present, Fig. 3), indicating that truncated 1/8-NORAD was excluded from SGs and only unbound probe localized to SGs, as confirmed by a side-by-side quantitative analysis of relevant ROIs (Supplementary Fig. 17). Higher cytosolic lifetime values (Fig. 6) suggest that 1/8-NORAD-Ribo(4A)-590 indeed remains in the cytosol. Quantification of differential subcellular NORAD partitioning demonstrates that Riboglow-FLIM reliably reports on RNA localization.

### Visualizing two RNAs with defined localizations by FLIM

Lastly, we visualized ACTB-Ribo(4D)-590 mRNA and 1/8-NORAD-Ribo(4A)-590 RNA simultaneously in the same cell, after inducing SGs (Fig. 7). The lifetime in ROIs corresponding to SGs was markedly lower than the cytosolic ROIs and close to the D-tag value, as expected for partitioning of ACTB-Ribo(4D)-590 to SGs, but not 1/8-NORAD-Ribo(4A)-590. In contrast, the fluorescence lifetime in the cytosol was indicative of a ~50:50 mix of ACTB-Ribo(4D)-590 and 1/8-NORAD-Ribo(4A)-590 in double-transfected cells (Fig. 7).

### Discussion

Together, we find that Riboglow-FLIM allows for quantitative live RNA visualization with key advantages over intensity-based approaches

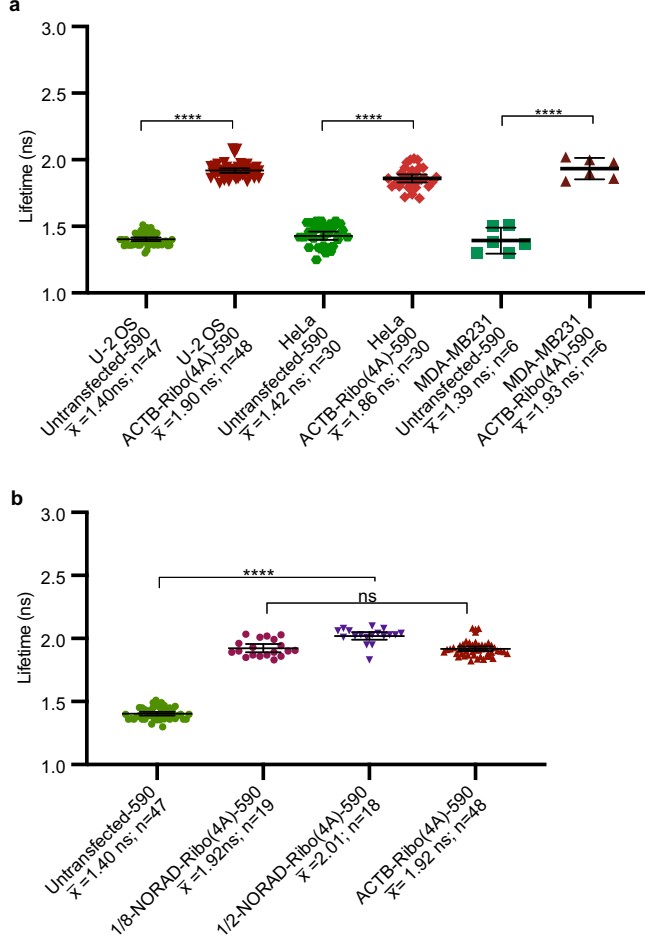

**Fig. 4 | Assessing Riboglow-FLIM in different cell lines and fused to variable RNAs.** Quantitative visualization of RNA sensing by fluorescence lifetime imaging microscopy (FLIM) for different live mammalian cell lines (**a**) and different RNAs tagged with Riboglow (**b**). Average fluorescence lifetime values (as defined in Fig. 2) for live cells (204 cells, 4 independent experiments) transfected with listed plasmids, identified via a transfection marker, loaded with Cbl-4xGly-ATTO590, and data processed as outlined in the workflow in Fig. 2a. One symbol = 1 cell, *p*-values listed (ns: $p \leq 0.5$; *$p \leq 0.05$; **$p \leq 0.01$; ***$p \leq 0.001$; ****$p \leq 0.0001$). One-way ANOVA (95% confidence limit); post hoc test (Tukey HSD). Error bars indicate mean and standard deviation (+/−SD).

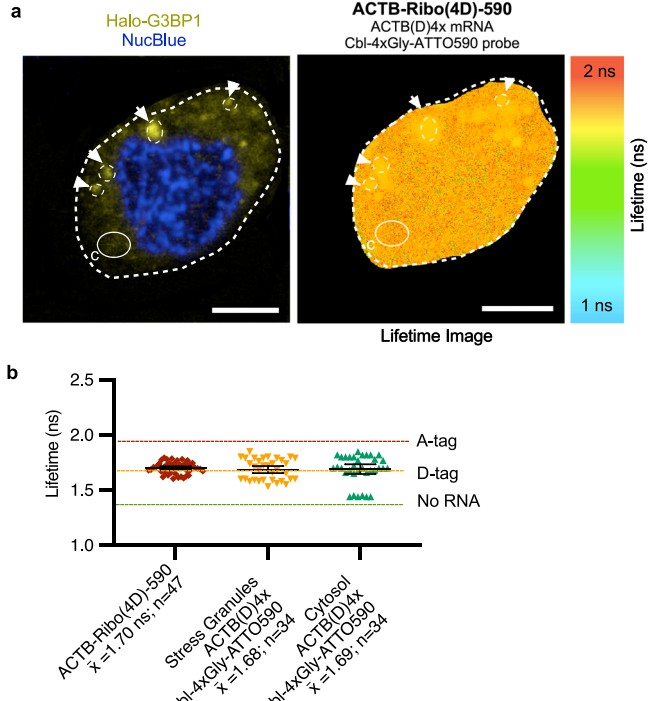

**Fig. 5 | Visualizing RNA recruitment to stress granules (SGs) live via Riboglow-tagging and detection by FLIM. a** Representative cell of arsenite-stressed Halo-Tag-G3BP1 U-2 OS cells transfected with ACTB-Ribo(4D)−590. Cells were co-transfected with a fluorescent transfection marker to identify which cells had taken up the plasmid encoding for the Riboglow-tagged reporter, Riboglow probe was loaded into live cells. The chromosomally encoded SG marker protein G3BP1 tagged with HaloTag was labeled with the HaloTag ligand JF646 live. Left: Fluorescence intensity image to visualize nucleus (blue) and SGs (yellow); white arrow = G3BP1-labeled SGs (scale bar = 10 μm). Right: Pixel-by-pixel component lifetime image. Regions of interest (ROI) of SGs identified via G3BP1 in the intensity image (left) were labeled in the lifetime image (right) for further analysis. **b** The average fluorescence lifetime value was determined for the whole cell and individual ROIs corresponding to each SG. White arrows pointing to circles in panel **a** and a randomly selected ROI of similar area in the cytosol (labeled with "C" in panel **a**) illustrate the ROIs for dot plot analysis (65 cells, 115 ROIs, 5 total independent experiments). Dotted lines represent the mean of the lifetime for benchmarks established from ACTB tagged mRNA in Fig. 3. One-way ANOVA (95% confidence limit); post hoc test (Tukey HSD). Error bars indicate mean and standard deviation (+/−SD).

(Supplementary Table 1). The small RNA tag size (~100 nt) does not impair cellular contrast. The differences in fluorescence lifetime for Riboglow tags with the same fluorescent probe are readily detectable and allow for visualizing orthogonal RNAs simultaneously, a goal not previously achieved with aptamer-based platforms. We anticipate that Riboglow-FLIM will expand multiplexing capabilities for RNA visualization in live cells.

## Methods

### Preparation of Riboglow Fluorescent Probe
Riboglow probes were a gift from Amy Palmer at CU Boulder and brought up in phosphate buffered saline (PBS)[11,21]. The concentration of Cbl-4xGly-ATTO 590 was determined using a published extinction coefficient ($120,000^{-1}$ mol$^{-1}$ at 594 nm). Stocks (5 μM) in PBS were prepared for live cell imaging and stored in the dark at −20 °C. A summary of Riboglow FISH probes is provided in Supplementary Data 2.

### Instrument Response Function (IRF)
An aliquot of 1000 μL supersaturated solution of potassium iodide (KI) and Rhodamine B was placed in a sterile 35 mm μ-dish with a polymer

cover slip (Ibidi). Imaging was conducted on an Abberrior STEDYCON microscope (excitation laser line at 594 nm) at high excitation power and the fastest acquisition setting, using a fixed 512 × 512 pixel area. SymPhoTime 64 (Picoquant) was used to extract the decay curve by measuring the fluorescence readout for 30 seconds, with care taken to avoid saturation effects. Picoquant instructions for IRF extraction were followed.

### DNA Preparation
All DNA samples were prepared following the basic Qiagen midi-prep procedure and diluted to 1 μg/μL in 1X TE (10 mM Tris-Cl, pH 8.0, 1 mM EDTA) buffer. Plasmid pCMV-GFP was a gift from Connie Cepko (Addgene plasmid # 11153)[22], pmCherry Stop is pmCherry-C1 (Clontech), GFP-Rab5B was a gift from Gia Voeltz (Addgene plasmid # 61802)[23], pEGFP-C1-G3BP1-WT (Addgene plasmid # 135997)[24] was a gift from Anthony Leung, ACTB-(A)4x was a gift from Amy Palmer (Addgene plasmid # 112058), ACTB-(A)1x was a gift from Amy Palmer (Addgene plasmid # 112055). ACTB-(D)4x, 1/2-NORAD-(A)4x and 1/8-NORAD-(A)4x were constructed using standard cloning procedures from pRP877 as follows. pRP877 was a gift from Roy Parker. NORAD-

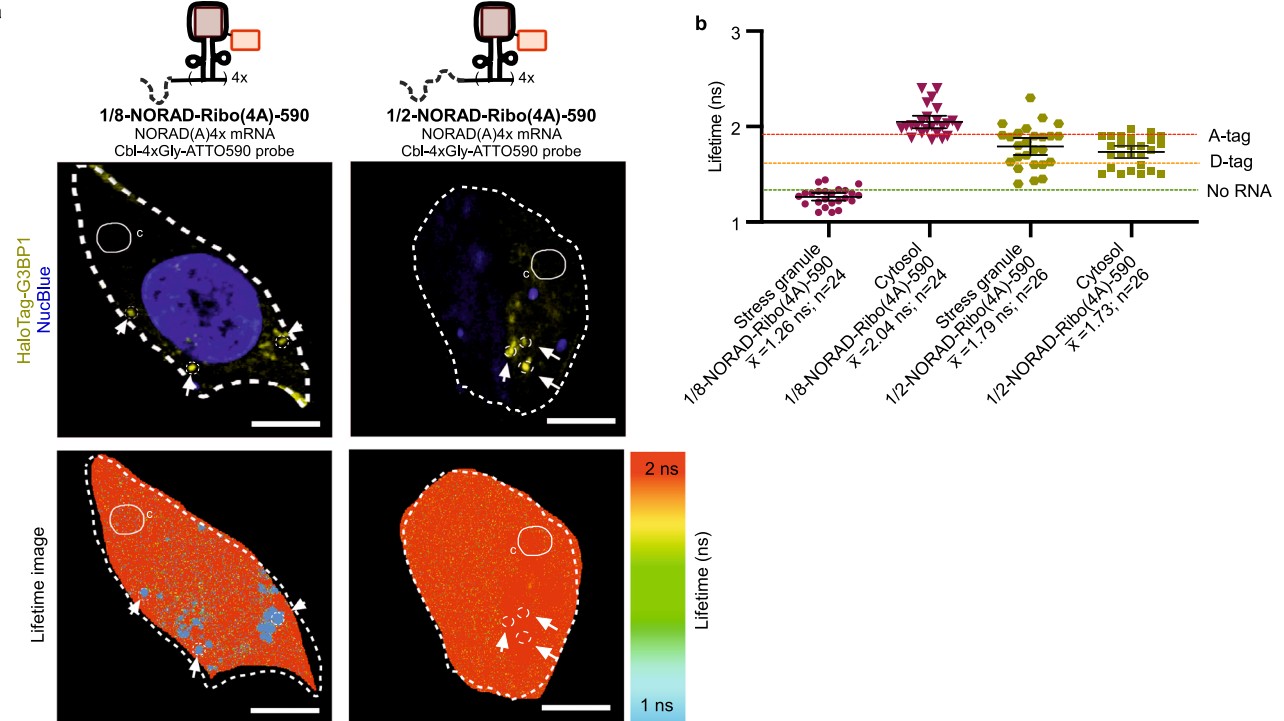

**Fig. 6 | Quantification of RNA recruitment to stress granules (SGs) for different tagged RNAs via Riboglow-FLIM. a** Representative images of arsenite-stressed HaloTag-G3BP1 U-2 OS cells producing truncated versions of Riboglow-tagged NORAD. Cells produce the SG marker protein G3BP1 from the chromosome to identify SGs via HaloTag ligand JF646 (top panel, fluorescence intensity image, yellow areas indicate G3BP1). Nucleus was identified using NucBlue (blue). White arrows point to HaloTag-G3BP1 marked SGs. Smaller white circle labeled C indicates a representative ROI of the cytosol. Bottom panel: pixel-by-pixel component lifetime image of the same field of view visualized on a false color scale. Scale bar = 10 μm. **b** The average fluorescence lifetime value was determined where each dot corresponds to an ROI representing the whole cell, or a SG (circles, indicated by white arrow), or a randomly selected ROI of similar area in the cytosol ('C') (23 cells, 100 ROIs, 8 total independent experiments, 1 symbol = 1 ROI). Dotted lines represent the mean of the lifetime for benchmarks established from ACTB tagged mRNA in Fig. 3. One-way ANOVA (95% confidence limit); post hoc test (Tukey HSD). Error bars indicate mean and standard deviation (+/−SD).

Riboglow truncations were built by replacing the ACTB sequence in plasmid ACTB-(A)4x with the corresponding NORAD truncation sequence from pRP877, resulting in plasmids 1/2-NORAD-(A)4x and 1/8-NORAD-(A)4x (see NORAD sequences in Supplementary Data 1, ACTB sequences in[11,21]).

## Fluorescence Lifetime Imaging Microscopy (FLIM)

Following incubation, the fluorescent probe was loaded into live cells as described below. A single imaging dish of cells was imaged for 2–3 hours using the Abberior STED FLIM microscope with a fixed imaging area of 512 × 512 pixels. Data was acquired using a Picoquant Timeharp 260 card. Data per frame was acquired until a total threshold of $10^4$ photon counts was reached with a pulsed laser of 40 MHz and excitation at 590 nm and adjusted to avoid photobleaching or photon pileup. Data was generated using Picoquant SymPhoTime 64 software. A false-color scale of FLIM images was set based on a range of lifetime histograms for measured samples and the average amplitude weighted lifetime images were extracted, as detailed below (for example Fig. 2b). An IRF standard was collected at the same time so that no subtraction of a background signal was necessary. We compared the lifetime values for the ROIs across conditions to ensure that no artifactse were introduced.

## Mammalian cell culture

Adherent U-2 OS, HeLa, and MDA-MB231 cells were obtained from the Tissue Culture and Biobanking Shared Resource (Georgetown University). U-2 OS Halo-G3BP1 cells[11] were a gift from Roy Parker (CU Boulder). Cells were passaged for up to 5 passages at 37 °C and 5% $CO_2$ in 10% FBS (Gibco) and Dulbecco's modified eagle medium (DMEM, Gibco). These cell lines are not listed as commonly misidentified by the International Cell Line Authentication Committee.

For microscopy experiments, cells were seeded at $0.25 \times 10^6$ cells in 10% FBS (Gibco) and Dulbecco's modified eagle medium (DMEM, Gibco) in sterile 35 mm μ-dishes with a polymer coverslip (Ibidi) and incubated at 37°C and 5% $CO_2$. Cells were transfected in imaging dishes following manufacturer's recommendations with TransIT®-2020. Briefly, cells were transfected with 1 μL (concentration of 1 μg/μL) of plasmid DNA stocks of interest, 6 μL TransIT®-2020 (Mirus), and 250 μL Opti-MEM (Thermo Fisher) following the manufacturer-recommended TransIT®-2020 protocol. Fresh media was supplied to cells prior to transfection. Plates were incubated for 24 - 72 hours and transitioned to FluoroBrite™ DMEM (Gibco) before further treatment (arsenite treatment, probe bead loading, addition of dyes) and imaging.

The Riboglow fluorescent probe (Cbl-4xGly-ATTO 590) was loaded into live cells as previously described[11,21] using a homemade bead loader. Briefly, the cell culture media was removed from imaging dishes, 3 μL of a 5 μM stock of the probe was added and loaded and fresh media was immediately added. Cells were incubated for 10 min at 37 °C and 5% $CO_2$ in 10% FBS (Gibco) and Dulbecco's modified eagle medium (DMEM, Gibco) and imaged within 3 hours of probe loading.

## Stress Granule (SG) media preparation and assay

U-2 OS Halo-G3BP1 cells were seeded in 35 mm imaging dishes (Ibidi) to a seeding density of $0.25 \times 10^6$ cells. 24 hours after seeding, cells were transfected with 1 μL (concentration of 1 μg/μL) of plasmid DNA stocks of interest, 6 μL TransIT®-2020 (Mirus), and 250 μL Opti-MEM (Thermo Fisher) following the TransIT®-2020 protocol. Janelia Fluor 646 ($JF_{646}$) dye (Promega) was prepared by diluting the JF dye to a concentration of 1 μM in 1 mL culture media (10% FBS/DMEM). 24–48 hours after transfection, the JF dye supplemented media was

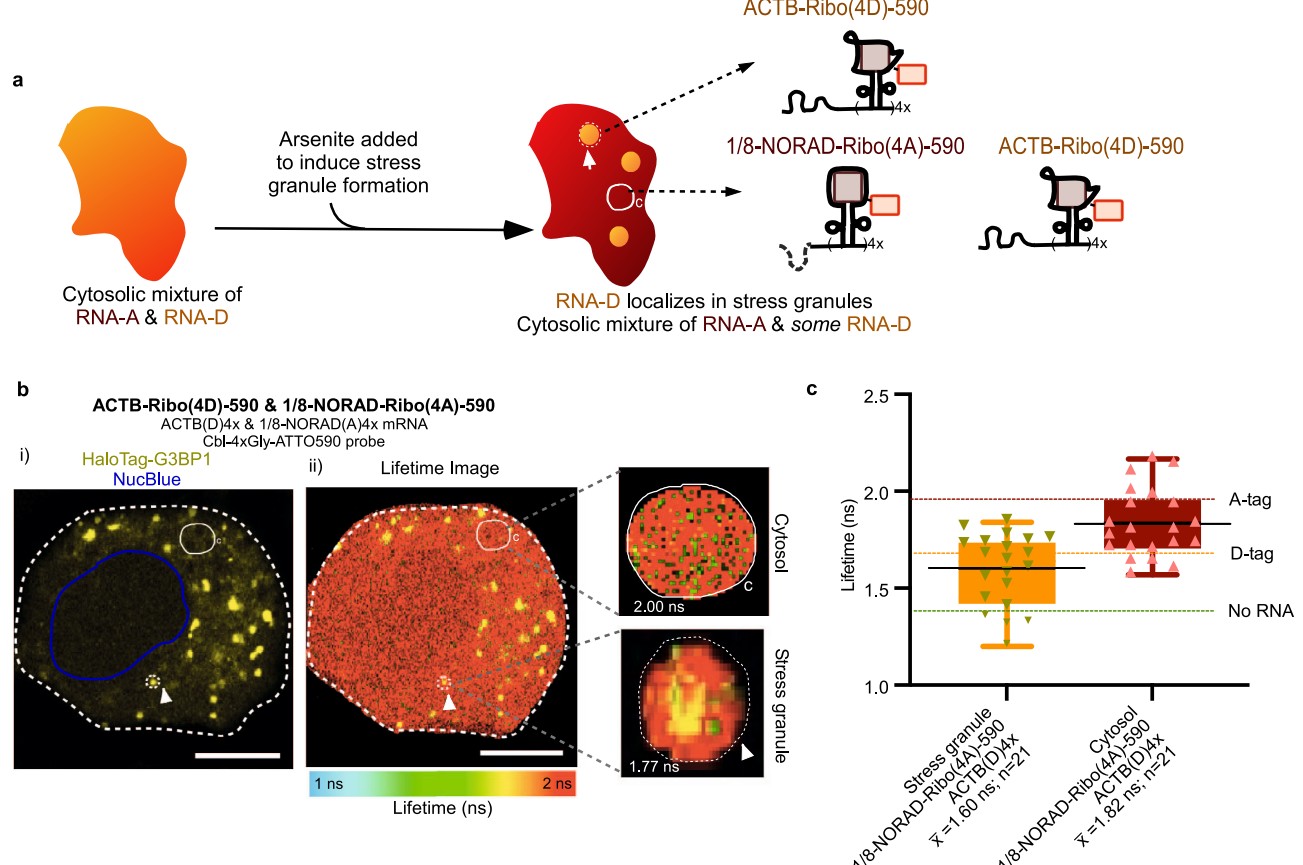

**Fig. 7 | Differential recruitment of two different model RNAs tagged with Riboglow-FLIM to stress granules (SG) in live cells. a** Cartoon illustration of simultaneously detecting subcellular localization of two RNAs via Riboglow RNA tags (A, D) that exhibit distinct lifetime values. **b** Representative images of arsenite-stressed HaloTag-G3BP1 U-2 OS cells simultaneously producing ACTB-Ribo(4D)−590 and 1/8-NORAD-Ribo(4A)−590. The Riboglow-tagged reporter was transfected together with a transfection marker, JF646 dye was added to mark the SG marker protein HaloTag-G3BP1, and the nucleus was labeled via NucBlue. Left: fluorescence intensity image showing SG and nucleus, and illustrating how SGs were identified (JF646-labeled SGs, yellow). Right: pixel-by-pixel component lifetime image. Dashed line with white arrow: ROI of SG. Solid line labelled with 'C': representative ROI of cytosol. Scale bar = 10 μm. **c** Average fluorescence lifetime (as defined in Fig. 2) value of ROIs representing SGs and cytosol, defined as an average value for the entire ROI, as illustrated in **b**. Lifetime values for live cells (21 cells, 42 ROIs, 2 independent experiments) listed in the box-whisker plot in which whiskers = minimum and maximum values, black line= mean, box represents interquartile range (25th percentile to 75th percentile). Dotted lines represent the mean of the lifetime for benchmarks established from ACTB-tagged mRNA in Fig. 4.

added to the seeded U-2 OS Halo-G3BP1, and plates were incubated for 20 minutes at 37 °C/5% CO₂. Cells were then washed with PBS to remove any unbound dye. Cells were bead loaded as outlined above. To minimize cell agitation and timing of cell prep, sodium(meta) arsenite (≥90%, S7400, Sigma-Aldrich) was added to the imaging media (FluoroBrite DMEM/10%FBS), yielding a final concentration of 0.5 mM sodium(meta)arsenite media, which was added directly to cells following the PBS washes after bead loading. Cells were incubated for 20 min at 37 °C/5% CO₂ to allow for stress granule formation (for example Fig. 5). Regions of interest (ROI) of both cytosol and stress granules were selected through inspection of Halo-G3BP1 granule formation (Supplementary Fig. 12). ROI data extraction of SGs was done sequentially, yielding a final image showing multiple ROI's in the same cell (for example Fig. 5).

To evaluate the effect of cellular RNA levels, cells were prepared as outlined above and the concentration of 1 μg/μL plasmid DNA stocks varied to be 0.1 μg/μL and 1 μg/μL.

**Multiexponential reconvolution fitting analysis**
Fluorescence decay curves obtained by FLIM were analyzed with reconvolution using SymPhoTime 64 reconvolution script (Picoquant) and the acquired IRF. Time-correlated single photon counting (TCSPC) measurements were conducted for live mammalian cells containing plasmids producing RNA reporters and chemical probes listed in

Supplementary Table 2. The photon arrival time at each pixel was summarized into a histogram of arrival times for a region of interest (ROI), defined here as an individual cell (Supplementary Fig. 18).

Details about different options for FLIM data analysis are discussed in Supplementary Note 1, where we conclude that multi-exponential reconvolution fitting is the desired data analysis for this study. Therefore, the acquired decay function at each pixel was analyzed further to extract fluorescence lifetime values through a multi-exponential reconvolution fit. The number of exponentials extracted from decay functions was assessed, see Supplementary Fig. 5 for examples. Together, the fit was assessed by evaluating (i) the overlay of the fitted curve over the decay curve, (ii) a random distribution of residuals, and (iii) the lowest number of parameters. Based on this, a tri-exponential reconvolution fit for cells that did not contain tagged RNA and a bi-exponential reconvolution fit for cells that did contain tagged RNA were found to be the most appropriate (Supplementary Fig. 5). All data was extracted and fitted using Symphotime64 reconvolution global analysis (Picoquant) and the resulting amplitude-weighted average lifetime was assigned to a false color scale for visualization (for example Fig. 2b).

**In vitro fluorescence lifetime measurements**
Fluorescence lifetimes were measured for samples containing Cbl-4xGly-ATTO590 probe, ATTO590-Biotin (Sigma), or free ATTO590

(Sigma) in the presence and absence of tagged RNA (A tag or D tag) (Supplementary Fig. 2). All experiments were conducted in RNA buffer (100 mM KCl, 1 mM MgCl$_2$, 10 mM NaCl, 50 mM HEPES, pH 8) with 5 μM of purified RNA and a final concentration of probe at 0.5 μM. Samples containing RNA were incubated for 20–30 minutes at room temperature to allow binding of the probe to RNA to occur prior. Data was acquired using the Abberrior STED FLIM microscope with a fixed imaging area of 512 × 512 pixels using a Picoquant Timeharp 260 card. Data per frame was acquired until a total threshold of 10$^4$ photon counts was reached with a pulsed laser of 40 MHz and excitation at 590 nm and adjusted to avoid photobleaching or photon pileup. Data was generated using Picoquant SymPhoTime 64 software and fit using multiexponential reconvolution fitting as described above with IRF consideration. A false-color scale for the FLIM images was set to 0 ns to 3 ns illustrating the different lifetimes visually. The intensity and amplitude-weighted lifetime values are reported and a representative lifetime image shown for samples containing (i) Cbl-4xGly-ATTO 590, (ii) Cbl-4xGly-ATTO 590 and A-tag and (iii) Cbl-4xGly-ATTO 590 and D-tag (Supplementary Fig. 2).

### RNA purification
DNA templates used for in vitro transcription were subcloned into pUC19 that included a T7 RNA polymerase promoter sequence. DNA templates were PCR amplified with Q5 High Fidelity protocol (NEB) and transcribed by T7 RNA polymerase using T7 High Yield RNA Synthesis Kit (NEB). Monarch® PCR & DNA Cleanup kits were used and samples run on agarose gel electrophoresis to confirm products with a 1 kb Plus Ladder (NEB). Monarch® RNA Cleanup Kit was used and samples run on a Urea-TBE precast gel (Biorad) to confirm transcription. RNA concentration was determined on a BioTek Synergy H1 Microplate Reader using RNA nanodrop capabilities on a Take3 Multi-Volume plate. Sequences of all RNAs used in this study are listed in Supplementary Table 2.

### Fluorescence in situ hybridization (FISH)
FISH using Human ACTB, Ribo-A tag, Ribo-D tag, and NORAD probes was performed guided by experiments previously[20,25]. Ribo-A tag and Ribo-D tag FISH probes were designed and produced by Integrated DNA Technologies (IDT) as standard DNA oligonucleotides with Cy5 conjugated through 5′ amino modifications. Each probe set contained 2-4 DNA oligonucleotides that were carefully designed to ensure RNA hybridization would occur. GC content was kept to ~40% ensuring minimal nonspecific binding. HPLC Purification was conducted and samples normalized to 100 μM in IDTE buffer pH 8.0. U-2 OS cells were cultured and treated as outlined before in 35 mm μ-dishes (Ibidi). Cells were fixed with 3.7% paraformaldehyde in 1X PBS for 10 minutes. Cells were permeabilized using 70% ethanol for at least 60 minutes at 4 °C and treated as described in the manufacturer's protocol (Stellaris®) with appropriate FISH probes (Stellaris® FISH probes Human ACTB with Quasar® 570 Dye, Stellaris® FISH probes Human ACTB with Quasar® 670 Dye, Stellaris® NORAD RNA with Quasar® 670, Stellaris® NORAD RNA with Quasar® 570, Ribo-A tag, Ribo-D tag, Supplementary Data 2). An Abberrior STEDYCON microscope was used for imaging with a fixed imaging area of 512 × 512 pixels and analyzed using STEDYCON smart control.

### Reporting summary
Further information on research design is available in the Nature Portfolio Reporting Summary linked to this article.

## Data availability
The data supporting the findings of this study are available from the corresponding authors upon reasonable request. The FLIM data generated in this study have been deposited in the FigShare database (https://doi.org/10.6084/m9.figshare.21445665.v1).

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

## Acknowledgements
The authors would like to acknowledge financial support from the NIH (R00GM127752, to E.B.), the Luce Foundation (to E.B.), and Georgetown College. S.R. is funded by the Department of Biochemistry and Cellular & Molecular Biology at Georgetown University as a full-time assistant professor, partially funded by NIH grant R01DK127830, and is a part of the Microscopy and Imaging Shared Resource (MISR), which is partially supported by NIH/NCI grant P30-CA051008. We thank P.L. Clark, R. Maillard, P. Roepe, R. Weiss, T. Stasevich, E. Vietmeyer, and S. Nair for helpful discussions. We thank the Tissue Culture and Biobanking Shared Resource and the Microscopy & Imaging Shared Resource (MISR) in the Georgetown Lombardi Comprehensive Cancer Center for technical support. We thank S. Lu, D. Clagett, and members of the Maillard group for technical assistance. We thank Amy Palmer for providing Riboglow probes, Roy Parker for providing plasmid pRP877 and U-2 OS Halo-G3BP1 cells, Connie Cepko for providing the pCMV-GFP plasmid (Addgene plasmid # 11153), Gia Voeltz for providing the GFP-Rab5B plasmid (Addgene plasmid # 61802), Anthony Leung for providing pEGFP-C1-G3BP1-WT (Addgene plasmid #135997), Amy Palmer for providing the ACTB-(A)4x plasmid (Addgene plasmid # 112058) and ACTB-(A)1x plasmid (Addgene plasmid # 112055). We thank M. Rice, T. Rotermund, A. Hampton, J. Alvey, L. Bowden, T. Bowden, L. Shafik, and U. Shankar for technical assistance.

## Author contributions
E.B. and N.S. conceptualized and designed the study. N.S., E.M., T.O., and H.L. performed experiments and analyzed data. N.S. developed FLIM data fitting procedures, with input from all authors. S.R. performed phasor analysis. N.S. and E.B. wrote the manuscript with edits from all authors.

## Competing interests
The authors declare no competing interests.
