## [Peer Review File · Nature Communications]

Visualizing orthogonal RNAs simultaneously in live mammalian cells by fluorescence lifetime imaging microscopy (FLIM)REVIEWER COMMENTS

Reviewer #1 (Remarks to the Author):

To date, orthogonal aptamers to tag multiple RNAs in the same cell have not been reported. The simultaneous detection of two different target RNAs in live cells is therefore a major challenge in RNA imaging that several groups in the field are pursuing. In this manuscript, Sarfraz and coworkers describe the simultaneous detection of two RNA in live cells by utilising the Riboglow-based RNA tagging system and FLIM approach. While the readout is in a single channel, the authors can discriminate between two different populations of exogenously expressed target RNAs with two orthogonal RNA aptamers that result in different lifetimes of the probe. The authors suggested that this evaluation method indeed allows to distinguish two individual populations of target RNAs, and thus the method enables to detect multiplexed RNAs. However, the provided data are not enough for the main statement of “multiplex visualisation”, and additional support evidence would be required. While the topic is highly interesting, this manuscript falls short in several ways. For Nature Commun, one would have expected more cell types and target RNAs to demonstrate the technology's broad applicability and permanently transfected cells. The number of cells in many reported experiments is also relatively low. As it stands, the manuscript seems written for a more specialised audience.

Major points

1. As the authors note in their manuscript, overexpression artefacts might affect the conformation and stoichiometry of tags and, therefore, the observed lifetimes. However, this aspect is not sufficiently explored, and additional control experiments would be necessary.
2. Tagging RNAs (A or D) provide different lifetime values. But there seems to be a lack of several control experiments to characterize the probe's lifetime with the different RNAs and under different conditions to convince potential readers that the observed lifetime differences result from the different tags and not the different environments or relative overexpression conditions.
3. The expression levels amongst a population of transfected cells typically vary significantly and, in general, cannot be controlled with asynchronous cells. Therefore, the authors need to provide additional statistics and control experiments to unambiguously demonstrate that differences in expression levels do not result in significant differences in lifetimes.
4. Fig2 > Is the FLIM image masked to show only the lifetime in the SG and cytoplasmic ROI? The FLIM image across the cell should be shown. It is unclear whether cellular sub-compartments can be identified purely based on lifetime measurements.
5. Fig4 > Based on the data in Figure 4c, it is not clear to me how the authors distinguish between SG and Cytosolic RNAs. The distributions overlap, and it is evident that the average lifetimes are statistically different. However, additional control experiments and statistical analysis seem to be lacking.

Minor points

1. Supplementary figure 6, line 82 > A-tag and D-tag; are they ACTB containing tags?

Reviewer #2 (Remarks to the Author):

This manuscript presents a new RNA imaging platform based on the Riboglow aptamer and fluorescence lifetime microscopy (FLIM). Riboglow has been known to bind the fluorogenic probe (Atto590 conjugated Cobalamine) and increases its fluorescence, enabling visualization of the RNAs using conventional intensity-based fluorescence microscopes. In this study, the authors found that the binding of the Riboglow aptamer to the probe significantly changes its fluorescence lifetime and they used this feature to detect Riboglow-fused RNAs in live cells using FLIM. They also showed that two variants of Riboglow (A and D) resulted in different fluorescence lifetimes upon binding to the probe, enabling visualization of two RNAs simultaneously.

The experiments are well described, the concept of using light-up aptamers in combination with FLIM is novel, and I find this study exciting. However, more efforts are needed to characterize the system in detail and to demonstrate the robustness of this platform. Overall, I think that this work is of broad interest and will be an important advancement in the RNA imaging field after the following points are clarified.

1. The system should be thoroughly characterized in vitro. Lifetimes of the free probe (Atto590 conjugated Cobalamine) and the aptamer bound forms should be measured, for both Riboglow variants. Since the quencher (Cobalamine) unit binds the aptamer, authors should discuss why they observed different lifetimes for the aptamer bound and free form and even for different variants. What is the lifetime of Atto590 (not the version with a free carboxylic acid group, but with the amide group as in Atto590-biotin)? Does that suggest possible interactions between the fluorophore and the aptamer? Does that suggest the presence of FRET between Cobalamine and Atto590 in the aptamer bound or free forms? How would contact quenching (in the free probe) and FRET affect the lifetime? Does Atto590 conjugated Cobalamine display the same fluorescence lifetime in the nucleus and the cytoplasm? I believe that these measurements and discussions are necessary to fully comprehend the Riboglow-FLIM imaging platform.

2. The authors should first optimize their Riboglow-FLIM imaging platform in live cells where fluorescence intensity-based imaging works indisputably. And then they should compare the image qualities in intensity-based and FLIM-based methods. Currently, the fluorescence intensity-based images in Figure 1b do not seem quite convincing and it is questionable that the authors were specifically able to image beta-actin mRNA (also see my comment #3). The fluorescence intensity in the control cell (i) is quite comparable to the others (ii, iii, iv) and the fluorescence signal in the nucleus is remarkably high. I suggest that the authors should express Riboglow (both variants) using the Tornado system (for cytosolic localization) and/or as a U6 RNA fusion (for nuclear localization) and then compare the intensity-based and FLIM-based images. After this, they should continue imaging more challenging systems such as beta-actin mRNA and NORAD. Currently, as a model system, the authors used mCherry expressing cells (SI Fig. 1), which is hardly similar to the Riboglow-FLIM platform.

3. FLIM images in Figure 1b do not seem to correctly report the localization of beta-actin mRNA. This

mRNA was shown to be mostly cytoplasmic in numerous studies. What is the reason for this inconsistency? Is it because of image processing? Is it because of bead loading? Is it because of overexpression of mRNA? Is it because RiboGlow changes the localization of the mRNA? These points should be clarified, and supported by FISH images. Authors should report a FLIM image where beta-Actin mRNA is correctly localized as in wild-type cell lines.

4. Localization and aggregation of NORAD constructs (with and without Riboglow tag) should also be confirmed by FISH.

5. I did not fully understand what kind of background subtraction did the authors apply to the FLIM images (Fig 1a, Fig2a, Fig3a). However, I would like to see even the free dye background (similar to Fig4b), rather than just plain black color inside the cells. They should use an appropriate LUT for display purposes. The selected regions in the images (Fig 1a, Fig2a, Fig3a) do not even have different shades. Why? It is hard to believe that pixel-by-pixel lifetimes are that homogeneous. The authors should always display pixel-by-pixel lifetimes inside the whole cell, not only in the aggregates.

6. Since the authors used only two variants of Riboglow (A and D) for imaging two distinct RNAs, it may not be appropriate to write “multiplexed visualization of RNA” in the title and the manuscript. If they create a library of Riboglow mutants with different lifetimes, then the current title would be more appropriate.

7. Figure 3: Why is the fluorescence lifetime inside the blue circles even lower than the lifetime of the free probe?

Reviewer #3 (Remarks to the Author):

The paper titled 'Multiplexed visualization of RNA dynamics in live mammalian cells by fluorescence lifetime imaging microscopy (FLIM) by Sarfraz et al demonstrates the capacity of FLIM to quantitatively image a novel RNA biosensor called Riboglow. This resource has the potential to be very useful for quantification of RNA localisation with respect to sub-cellular compartments, but additional experimental detail and controls experiments would strengthen this application.

Major Comments

1. Riboglow is dim in the RNA unbound state and fluorescent in the RNA bound state and the hypothesis is that this probe's lifetime will increase upon RNA binding. In Fig 1 it is demonstrated that for a given Riboglow construct the average fluorescence lifetime is homogeneous throughout the cell. Is the expectation that the fraction of RNA bound / unbound in each pixel is equal throughout the cell?

2. In Fig. 2 it would be useful to see an intensity image of the HaloTag-G33BP1 and NucBlue signal before arsenite stress (like in Fig S5) alongside an unmasked intensity and lifetime map of ACTB-Ribo(4D)-590,

and then the stressed example with inclusion of the un-masked intensity and lifetime image of ACTB-Ribo(4D)-590, to assess if intracellular variation exists prior to stress, and assess the impact of these additional fluorescent labels on the lifetime map of ACTB-Ribo(4D)-590. Especially given the potential for FRET as explained in the next comment.

1. In Fig. 2-4 stress granules are labelled with Halo-G3BP1-JF646, which has an excitation spectrum that overlaps with the emission spectrum of ACTB-Ribo(4D)-590 or 1/8-NORAD-Ribo(4D)-590. There is therefore a potential for a FRET interaction between these constructs that would lead to a quenching of the ACTB-Ribo(4D)-590 or 1/8-NORAD-Ribo(4D)-590 lifetime at stress granules. Was this potential artefact eliminated with respect to Fig 4 for example, where a shortened lifetime is indeed observed at stress granules? If not it should be demonstrated that FRET does not impact the measurement.

Minor Comments

1. I found the level of detail provided with respect to the methodology to be limited and sometimes difficult to understand. In many cases this detail is embedded in the supplementary information, but it would be good to provide these details in some capacity in the main manuscript to enable better understanding. For example, from the outset of the results, it would be good to state that the probe fluorophore used is Atto590 and clarify what is the purpose of the mCh protein control experiment for FLIM data processing, given that the IRF was calibrated with Rho B.

4. I found the results presented with respect to Fig 1 confusing since sample i versus iv are described first then sample iii and then finally ii. Potentially it would make more sense to either explain what all four samples are being analysed from the outset (i, ii, iii, iv), then explain in order of presentation, or rearrange figure one such that the sample order is i, iv, iii, ii. Note, the non-significant result described with respect to ii vs iv in Fig. 1c is not labelled in the plot.

During revision of this manuscript, all figure numbers were updated from the initial submission. Because the reviewer comments required substantial text edits and figure rearrangements, we chose to highlight key text changes by color, rather than using track changes. We color coded changes to simplify evaluating the revised manuscript and incorporation of the changes as follows:

Green: No new data

Pink: No new data, updated analysis as requested by reviewers.

Blue: New data / text, as requested by reviewers.

Yellow: Additional data to improve statistics as requested by reviewers.

Reviewer #1 (Remarks to the Author):

To date, orthogonal aptamers to tag multiple RNAs in the same cell have not been reported. The simultaneous detection of two different target RNAs in live cells is therefore a major challenge in RNA imaging that several groups in the field are pursuing. In this manuscript, Sarfraz and coworkers describe the simultaneous detection of two RNA in live cells by utilising the Riboglow-based RNA tagging system and FLIM approach. While the readout is in a single channel, the authors can discriminate between two different populations of exogenously expressed target RNAs with two orthogonal RNA aptamers that result in different lifetimes of the probe. The authors suggested that this evaluation method indeed allows to distinguish two individual populations of target RNAs, and thus the method enables to detect multiplexed RNAs. However, the provided data are not enough for the main statement of “multiplex visualisation”, and additional support evidence would be required. While the topic is highly interesting, this manuscript falls short in several ways. For Nature Commun, one would have expected more cell types and target RNAs to demonstrate the technology's broad applicability and permanently transfected cells.

We thank the reviewer for this comment. We indeed find that detecting two RNAs simultaneously is a major strength of this study. We acknowledge that “multiplexing” may require more than two orthogonal RNA tags and have changed the manuscript title and text to remove “multiplexing” and use “orthogonality of two RNAs” instead.

We agree that broad applicability requires various cell types. In addition to RNA visualizing in U-2 OS cells, we added two more cell types, HeLa cells and MDA-MB231 cells and determined that RNA detection by FLIM appears largely independent of cell system used (Fig. 4a, Results and Discussion paragraph 8).

While more target RNAs may be interesting to tag, we believe that using a well-established mRNA tag (ACTB) and a non-coding RNA (NORAD) are representative of possible applications of this tool. We find that these two RNAs represent breath of applicability: ACTB mRNA is a common target for mRNA visualization and may serve as a benchmark across RNA visualization tools. Indeed, ACTB mRNA was used for the first Riboglow study as well

(Brasemann et al, 2018, NCB). NORAD represents a new RNA that is non-coding and was never tagged before. We demonstrate that the RNA FLIM readout is independent of the RNA that is tagged, the cell line used, subcellular localization, and stress (Fig. 4, 5, 6, Supp. Fig. 8, 15, 18). In light of this robust demonstration of Riboglow-FLIM as a tool we think that adding more RNAs beyond these would be beyond the scope of this study.

We agree that permanently transfected cells (i.e. producing the RNA from the chromosome) would be a valuable additional dataset. However, we understand that producing tagged RNAs from the chromosome appears to be challenging in the RNA imaging field at large. In fact, producing fluorescently tagged RNAs from the chromosome in mammalian cells has not been achieved for any other fluorescent RNA tag to date to our knowledge. We hypothesize that this is largely due to the resulting low RNA levels in live cells that are often not compatible with intensity-based visualization. Towards this goal, we have varied levels of the RNA by altering the concentration of transfected plasmid DNA (Supp. Fig. 15). As expected, live RNA detection by FLIM remains robust, independent of plasmid DNA levels. It is noteworthy that fluorescence intensity-based detection is much more heterogeneous and dependent on precise transfection conditions (Supp. Fig. 15). While we predict that Riboglow-FLIM may be compatible with visualizing tagged RNAs from the chromosome, visualizing RNAs produced from the chromosome is beyond the scope of this study.

The number of cells in many reported experiments is also relatively low.

We thank the reviewer for this comment. We have increased the numbers of cells (indicated in yellow in the legends for Fig. 3, 4, 5, 6, 7):

Experiment	Original sample number reported	Current sample numbers
Untransfected-590; ACTB-Ribo(4D)-590	75	94
Untransfected-590; ACTB-Ribo(4A)-590	76	167
Untransfected-590; 1/8-NORAD-Ribo(4A)-590; 1/2-NORAD-Ribo(4A)-590; ACTB-Ribo(4A)-590	90	132
Untransfected-590; ACTB-Ribo(4D)-590; ACTB-Ribo(4A)-590; ACTB-Ribo(1A)-590	140	160
ACTB-Ribo(4D)-590 Stress Granules	60	115
1/8-NORAD-Ribo(4A)-590; 1/2-NORAD-Ribo(4A)-590 Stress Granules	66	100
Untransfected-590; 0.1uL 1/2-NORAD-Ribo(4A)-590; 1uL 1/2-NORAD-Ribo(4A)-590	22	54

We would also like to add that the numbers of cells used in live cell FLIM studies are typically much lower than in our study in the field. We conclude that our additional data points are sufficient to have a robust dataset. A few recent examples of FLIM studies in the field are listed below.

Literature	Range of cells imaged
Summers, P.A., Lewis, B.W., Gonzalez-Garcia, J. et al. Visualising G-quadruplex DNA dynamics in live cells by fluorescence lifetime imaging microscopy. Nat Commun 12, 162 (2021).	6 - 20
Blacker, T., Mann, Z., Gale, J. et al. Separating NADH and NADPH fluorescence in live cells and tissues using FLIM. Nat Commun 5, 3936 (2014).	9 -17
Scipioni, L., Rossetta, A., Tedeschi, G. et al. Phasor S-FLIM: a new paradigm for fast and robust spectral fluorescence lifetime imaging. Nat Methods 18, 542–550 (2021).	28
Castello, M., Tortarolo, G., Buttafava, M. et al. A robust and versatile platform for image scanning microscopy enabling super-resolution FLIM. Nat Methods 16, 175–178 (2019).	10
Wallrabe, H., Svindrych, Z., Alam, S.R. et al. Segmented cell analyses to measure redox states of autofluorescent NAD(P)H, FAD & Trp in cancer cells by FLIM. Sci Rep 8, 79 (2018).	21

As it stands, the manuscript seems written for a more specialised audience.

We thank the reviewer for this comment. We have expanded the text to include broad applicability of the platform and explain FLIM and its strengths for live cell imaging more broadly to reach a wider audience. Details about our data collection and processing workflows were expanded, for example: Fig. 2 (flowchart to elaborate on data processing), Supplementary Fig. 12 (flowchart to illustrate SG assay), Results and Discussion paragraph 3, Methods section “Multiexponential deconvolution fitting analysis”, supported by Supplementary Note 1 and 2.

Major points

1. As the authors note in their manuscript, overexpression artefacts might affect the conformation and stoichiometry of tags and, therefore, the observed lifetimes. However, this aspect is not sufficiently explored, and additional control experiments would be necessary.

We thank the reviewer for this excellent comment. We agree that overexpression may induce artifacts and needs to be considered. We added additional control experiments to confirm that expressing our model RNAs (ACTB mRNA and NORAD, detected in cytosol and stress granules) from a plasmid with a Riboglow tag mimics localization as seen for endogenous, untagged versions (Supplementary Fig. 3 and 13, Results and Discussion paragraphs 2 and 9). We also want to point out that the localizations we observed are consistent with what was demonstrated in the literature. We emphasize that point more clearly in the text now (Results and Discussion paragraph 9,10).

As an additional control, we have systematically altered levels of the RNA in cells by varying concentrations of the transfected reporter. We consistently see that the lifetimes of the reporter are not affected, likely due to the independence from concentration. In contrast, fluorescence intensity measurements are highly variable (Supplementary Fig. 15, Results and Discussion paragraph 5, 11).

We also note that using just one copy of the RNA sequence (<100 nucleotides, 1xA) significantly reduces the possibility of RNA tag misfolding, probe binding stoichiometry issues, and fluorescence intensity signal heterogeneity (Fig. 3). While intensity artifacts are addressed by using fluorescence lifetime, the 1x tag further reduces potential artifacts.

2. Tagging RNAs (A or D) provide different lifetime values. But there seems to be a lack of several control experiments to characterize the probe's lifetime with the different RNAs and under different conditions to convince potential readers that the observed lifetime differences result from the different tags and not the different environments or relative overexpression conditions.

We thank the reviewer for this comment. We added control experiments and re-organized the text to highlight these important controls, as follows:

- We have added in vitro measurements to demonstrate that the A and D tag yield the same reported lifetime values where the probe was added to purified RNA, as in the complex cell environment (Supplementary Fig. 1, Results and Discussion paragraph 1).

- We performed fluorescence lifetime and intensity experiments for the same tagged RNA in the same cell system for both ACTB mRNA and NORAD (Fig. 4b, Results and Discussion paragraph 10), similar to our observation that the cell type does not affect fluorescence lifetime (Fig. 4a, Results and Discussion paragraph 10).

- Our assay to vary expression levels demonstrates that the lifetime of the A tag only marginally varies for different plasmid DNA levels as a tool to change RNA levels, in contrast to fluorescence intensity values (Supplementary Fig. 15).

- We have carefully assessed lifetime values in subcellular compartments. While we do see slight changes in lifetimes in the nucleus, we note that this change is independent of the presence of the RNA tag, and likely due to an artifact of the probe in the nucleus (Supplementary Fig. 8, Results and Discussion paragraph 5). We like to point out that the probe signal in the nucleus is much more pronounced in fluorescence intensity measurements than FLIM, strengthening the point that FLIM added robustness to the Riboglow system. Furthermore, the slight difference in nuclear FLIM signal is much smaller than the differences we observed for our experimental system (see Supplementary Fig. 8, compare untransfected and transfected samples).

- When comparing FLIM values in the cytosol vs. stress granules, no statistically different values were observed, pointing to robustness of FLIM (Supplementary Fig. 8).

3. The expression levels amongst a population of transfected cells typically vary significantly

and, in general, cannot be controlled with asynchronous cells. Therefore, the authors need to provide additional statistics and control experiments to unambiguously demonstrate that differences in expression levels do not result in significant differences in lifetimes.

We thank the reviewer for this excellent point. As noted above, we have added control experiments and statistical analysis to demonstrate that differences in cellular RNA levels only minimally affect lifetimes, and much less pronounced than fluorescence intensity (Supplementary Fig. 15, Results and Discussion paragraph 5, 11). However, as demonstrated with FLIM live cell imaging the lifetimes only slightly varied and were thus expression levels were considered to be minimally perturbing. To validate that transfecting with increased concentrations of the plasmid DNA can serve as an (indirect) tool to vary RNA levels, FISH experiments were conducted on cells transfected under the specified condition in Fig. 3. As expected, significantly higher levels of RNA expression were noted and granules formed in cells transfected with increased concentrations of DNA (Supplementary Fig. 15).

4. Fig2 > Is the FLIM image masked to show only the lifetime in the SG and cytoplasmic ROI? The FLIM image across the cell should be shown.

The original Fig. 2 has been updated (now Fig. 5). We also separated the original Fig. 1 (now Fig. 1, 2 and 3) and now include a flowchart to make our image analysis workflow more clear (Fig. 2a). Whole cell lifetime values can be reported in different ways. First, the average component lifetime of the whole cell may be reported, where the whole cell is defined as the region of interest. We use this analysis workflow to extract one lifetime value for our dot plot analysis (e.g. Fig. 5b). Alternatively, lifetimes can be resolved pixel-by-pixel. The other image shown is where the individual subcellular compartments are selected as the specified ROI (i.e., stress granules were identified by Halo-G3BP1 signal, first panel) and individually fit to extract a single representative lifetime value, namely the average lifetime value, and that average value is visually represented using a false color look up table. The latter has been selected as the primary form of representing FLIM data for ease of comparison between data sets using multiexponential deconvolution extracted average lifetime numbers that are assigned to an average lifetime value (i.e., dot plots in Fig. 3). Pixel-by-pixel lifetime visuals may be used to highlight subcellular lifetime heterogeneity (i.e., Fig. 5). In Fig. 2, we demonstrate both analysis option for the same cell to illustrate this. Text to elaborate on this point was added: Results and Discussion paragraph 3, Methods section “Multiexponential deconvolution fitting analysis”, supported by Supplementary Note 1 and 2.

It is unclear whether cellular sub-compartments can be identified purely based on lifetime measurements.

The reviewer is correct, and we were concerned to induce bias when assessing sub-compartmental localizations. As a solution, we used the protein marker G3BP1 to identify stress granules unambiguously and only analyze the SG localization based on the G3BP1 ROI identification. We made this approach more clear in the text (Results and Discussion paragraph 9), and added a flowchart figure to elaborate (Supplementary Fig. 12).

5. Fig4 > Based on the data in Figure 4c, it is not clear to me how the authors distinguish between SG and Cytosolic RNAs. The distributions overlap, and it is evident that the average

lifetimes are statistically different. However, additional control experiments and statistical analysis seem to be lacking.

See above – we use the protein G3BP1 with a fluorescent marker to identify SGs (Supplementary Fig. 11). Regions of interest (ROI) of both cytoplasm and stress granules were selected through inspection of Halo-G3BP1 granule formation and comparison of acquired images as shown in Fig. 5-7. We made this point clearer in the text (Results and Discussion paragraph 9) and added a flowchart figure to elaborate this (Supplementary Fig. 12).

Minor points

1. Supplementary figure 6, line 82 > A-tag and D-tag; are they ACTB containing tags?

This is now Fig. 4b. These were NORAD-tagged RNAs. To orient the reader to our lifetime values that correspond to untransfected cells with the probe only vs. those that also produce the Riboglow A-tag or D-tag, we added dotted lines (Fig. 4b). We clarified the figure legend to reflect that the lifetime mean lines represent the benchmark established lifetimes from Figure 4.

Reviewer #2 (Remarks to the Author):

This manuscript presents a new RNA imaging platform based on the Riboglow aptamer and fluorescence lifetime microscopy (FLIM). Riboglow has been known to bind the fluorogenic probe (Atto590 conjugated Cobalamine) and increases its fluorescence, enabling visualization of the RNAs using conventional intensity-based fluorescence microscopes. In this study, the authors found that the binding of the Riboglow aptamer to the probe significantly changes its fluorescence lifetime and they used this feature to detect Riboglow-fused RNAs in live cells using FLIM. They also showed that two variants of Riboglow (A and D) resulted in different fluorescence lifetimes upon binding to the probe, enabling visualization of two RNAs simultaneously.

The experiments are well described, the concept of using light-up aptamers in combination with FLIM is novel, and I find this study exciting. However, more efforts are needed to characterize the system in detail and to demonstrate the robustness of this platform. Overall, I think that this work is of broad interest and will be an important advancement in the RNA imaging field after the following points are clarified.

We thank the reviewer for this overall assessment. We added more data to ensure robustness and additional controls, summarized here.

- We increased our numbers of cells and independent experiments (amounting to at least 30 cells per condition) to ensure reproducibility and robustness across all data sets (summarized in table form above). Overall, comparable FLIM studies often report using far less cells per conditions (see above for details).

- *in vitro* FLIM experiments (Supplementary Fig. 1, 2, Results and Discussion paragraph 1) have been added demonstrating the relevant lifetime changes in the presence and absence of RNA tags inducing lifetime changes. This further highlights robustness of the system.

- FISH experiments were added to demonstrate the localization the RNA models that were used and compared to literature (Supplementary Fig. 3, 7, 13).

We would like to point out that the following analyses demonstrate robustness, especially for a new FLIM application:

- A blind run experiment was conducted to further demonstrate reproducibility (Supplementary Fig. 10).

- FLIM data sets in this paper were evaluated using numerous models of fitting and fit-free analysis (phasor) and demonstrated the same robust results independent of analysis workflow (Supplementary Table 3, Results and Discussion paragraph 6).

1. The system should be thoroughly characterized in vitro. Lifetimes of the free probe (Atto590 conjugated Cobalamine) and the aptamer bound forms should be measured, for both Riboglow variants.

We thank the reviewer for this comment. The in vitro characterization was added in which RNA tags (A and D) were purified and lifetime measurements acquired in the presence of probe (Cbl-4xGly-ATTO590). The results from the in vitro measurements are listed shown in Supplementary Figure 1 and are comparable to previously reported lifetimes of the tags (Brasemann et al, 2018, NCB), demonstrating the applicability of fluorescence lifetime as a quantitative tool for imaging and identification of RNA.

Since the quencher (Cobalamine) unit binds the aptamer, authors should discuss why they observed different lifetimes for the aptamer bound and free form and even for different variants.

We thank the reviewer for this comment. Text was added (Results and Discussions paragraph 1), and a reference to the original paper where this was first observed in vitro (Brasemann et al, NCB, 2018) was discussed in this context.

What is the lifetime of Atto590 (not the version with a free carboxylic acid group, but with the amide group as in Atto590-biotin)?

Average fluorescence lifetime values for in vitro measurements of ATTO590 probe variants (Cbl-4xGly-ATTO590, ATTO590-Biotin and free ATTO590) in the presence and absence of purified Riboglow RNA tag (A-tag) were measured and listed in Supplementary Figure 2. The lifetime differences were not statistically significant for the ATTO590-Biotin and free ATTO590 in the presence and absence of RNA tags. Whereas, Cbl-4xGly-ATTO590 has a significant increase in lifetime in the presence of the RNA tag thus demonstrating the high specificity to Cbl and the fluorescence turn on upon binding of probe bound to Cbl.

Does that suggest possible interactions between the fluorophore and the aptamer? Does that suggest the presence of FRET between Cobalamine and Atto590 in the aptamer bound or free forms? How would contact quenching (in the free probe) and FRET affect the lifetime?

We thank the reviewer for this interesting and relevant comment. The possible quenching mechanism (and consequences for FRET / lifetime) was discussed in the original Riboglow study (Brasemann et al, NBC, 2018). We would like to note that a mechanistic investigation may be the focus of a follow-up study.

Does Atto590 conjugated Cobalamine display the same fluorescence lifetime in the nucleus and the cytoplasm?

This important experiment was added. We do observe slight differences in fluorescence lifetime even for the probe alone when comparing lifetime in the cytosol and nucleus (in the absence of RNA) (Supplementary Fig. 8, Results and Discussion paragraph 5). This effect was also observed for fluorescence intensity, to a much greater extent (see for example Fig. 2, 3, and Supplementary Fig. 8). This was also observed for images in Braselmann et al, NCB, 2018. Indeed, we believe that this improvement of nuclear nonspecific signal is much improved for FLIM vs. intensity-based imaging.

We also want to point out that our model RNA tracking did not include nuclear RNAs to avoid potential artifacts from this effect, if there are any. A follow-up study will investigate causes of this effect to mitigate possible artifacts and establish Riboglow for visualizing dynamics in the nucleus.

I believe that these measurements and discussions are necessary to fully comprehend the Riboglow-FLIM imaging platform.

We thank the reviewer for these detailed and thoughtful comments. We believe we have added all suggested measurements and discussions.

2. The authors should first optimize their Riboglow-FLIM imaging platform in live cells where fluorescence intensity-based imaging works indisputably.

We agree with the reviewer that this experiment is an important prerequisite. Stress granules experiments using intensity-based imaging was performed in a prior publication (Braselmann et al, 2018, NCB). A clearer reference to this work was added (Results and Discussion paragraph 1) to the current manuscript. We also added control experiments in fixed cells by FISH (Supplementary Fig. 3, 7, 13) to serve as a basis for this study.

And then they should compare the image qualities in intensity-based and FLIM-based methods. Currently, the fluorescence intensity-based images in Figure 1b do not seem quite convincing and it is questionable that the authors were specifically able to image beta-actin mRNA (also see my comment #3).

We thank the reviewer for this important comment.

- We agree that careful controls to confirm that beta-actin mRNA was detected are required. We want to highlight that the same construct (ACTB-Ribo(4A)-590) was used in a prior publication (Braselmann et al, 2018, NCB) and extensive FISH control experiments were added (Supplementary Fig. 3, 7, 13). We acknowledge that the construct naming was altered, which makes it difficult to follow between publications.

- We also added FISH control experiments to confirm that beta-actin mRNA used in this study localizes as expected (Supplementary Fig. 3, 7, 13).

The fluorescence intensity in the control cell (i) is quite comparable to the others (ii, iii, iv) and the fluorescence signal in the nucleus is remarkably high.

We thank the reviewer for pointing out this important observation. Indeed, the fluorescence intensity between conditions (plus/minus RNA tag, RNA tag A vs. D) is quite similar (Fig. 3). However, differences in the lifetime signal are robustly different, a strength of this study in light of the rather subtle intensity value differences, in our opinion.

We agree that the signal in the nucleus is higher than the cytosol for all cells, including the control cells. This has been observed for intensity-based Riboglow imaging before (Brasemann et al, NCB, 2018). As mentioned above, we find that the slight increase in nuclear signal is much less pronounced for FLIM vs. intensity and represents a strength of the FLIM application (Supplementary Fig. 8). We find that the nuclear signal increase is not dependent on the RNA ligand and therefore not a result of RNA tag localization, as we see the same effect even in the absence of RNA (untransfected, Supplementary Fig. 8). The slight nuclear fluorescence signal increase will be the focus of a follow-up study.

I suggest that the authors should express Riboglow (both variants) using the Tornado system (for cytosolic localization) and/or as a U6 RNA fusion (for nuclear localization) and then compare the intensity-based and FLIM-based images.

We appreciate this thoughtful comment asking for a systematic evaluation of Riboglow-FLIM. We re-organized text and figures to help guide the reader through our systematic approach more clearly (Supplementary Fig. 12, Fig. 2). We also made clearer references to previously published work on the Riboglow system, especially work on Riboglow-tagged beta actin mRNA (Brasemann et al, 2018, NCB) (Results and Discussions, paragraph 1) that includes extensive control datasets. We further added additional control experiments to this study (Supplementary Fig. 1, 2, 3, 4, 7, 13, 14).

After repeatedly purchasing stocks of plasmid pAV-U6+27-Tornado-Broccoli from Addgene to perform the suggested Tornado control experiment, we spent 9 weeks unsuccessfully attempting to add the Riboglow tag to this plasmid. Ultimately, we realized that there appears to be a mismatch between the deposited Addgene Tornado plasmid (124360) and the documentation supplied to Addgene. Specifically, it appears that the plasmid we received from Addgene did not include the Tornado sequence. To not delay the resubmission further, we proceeded without the Tornado tagging, because we believe that our beta-actin mRNA system (established previously, Brasemann et al, 2018, NCB) serves as an appropriate control for cytosolic mRNA detection (see below). Hence, we do not feel that Tornado tagging is required to establish robust visualization of Riboglow live.

- Our ACTB mRNA tagging system serves as a cytosolic model system. We highlighted in the text (Results and Discussion paragraph 3) that this system was used previously (Brasemann et al, NCB, 2018) with extensive control experiments. We added additional controls in this study (Supplementary Fig. 7, 13).

- While U6 tagging serves as an important control in the field, we believe that using a nuclear RNA as a model for this study could add complications due to slight fluorescence lifetime signal differences between cytosol and nucleus (see also discussed above). While FLIM improved this artifact substantially compared with prior intensity-based imaging (Brasemann et al, NCB, 2018) (Supplementary Fig. 8), we are concerned that using a nuclear RNA as a model at this stage may add unnecessary complications due to the slight fluorescence lifetime signal increase. While FLIM reduces the nuclear effect substantially compared with intensity, we believe a

thorough investigation of slight increase in nuclear lifetime values is needed, but out of the scope of this study. We plan to focus on the nuclear signal systematically in a follow-up study.

- We find that using NORAD as a model for another tagged RNA serve the purpose of expanding the Riboglow FLIM system for tagging a model noncoding RNA.

After this, they should continue imaging more challenging systems such as beta-actin mRNA and NORAD.

We thank the reviewer for these thoughtful comments. We added comments in the text to highlight that our beta-actin mRNA system was already established for intensity-based imaging previously (Brasemann et al, 2018, NCB) (Introduction paragraph 3). The published beta-actin work with extensive control experiments serves as a basis for our FLIM study. We also added more control experiments to establish beta-actin mRNA imaging in this study. Additionally, more FISH control experiments to confirm correct NORAD localization were added (Supplementary Fig. 3, 7, 13).

Currently, as a model system, the authors used mCherry expressing cells (SI Fig. 1), which is hardly similar to the Riboglow-FLIM platform.

We agree that mCherry imaging is very different from RNA imaging. We use mCherry as a model to establish robust FLIM data collection and analysis, as we found that image processing protocols for FLIM often vary in the field and tend to be more complex than intensity-based fluorescence imaging. We find that lifetime values for mCherry in our hands are indistinguishable from values in the literature (Supplementary Fig. 4), confirming that our analysis for Riboglow-FLIM is robust.

3. FLIM images in Figure 1b do not seem to correctly report the localization of beta-actin mRNA. This mRNA was shown to be mostly cytoplasmic in numerous studies. What is the reason for this inconsistency? Is it because of image processing? Is it because of bead loading? Is it because of overexpression of mRNA? Is it because RiboGlow changes the localization of the mRNA? These points should be clarified, and supported by FISH images. Authors should report a FLIM image where beta-Actin mRNA is correctly localized as in wild-type cell lines.

We thank the reviewer for this important comment. The reason for the observed RNA signal distribution compared to what would be expected from FISH or intensity-based imaging is the FLIM data processing. We have added a figure to explain this in detail (Fig. 2). We now use pixel-by-pixel analysis for cell representations that more accurately reflects what would be observed by FISH or intensity-based imaging (Fig. 2, Results and Discussion paragraph 3). We also added FISH data to confirm endogenous beta-actin localization (Supplementary Fig. 3, 13). These controls confirm that the Riboglow tag do not alter beta-actin mRNA localization, as we have determined previously as well (Brasemann et al, NCB, 2018).

We have also added a control demonstrating that bead loading does not affect mRNA localization (Supplementary Fig. 7).

4. Localization and aggregation of NORAD constructs (with and without Riboglow tag) should also be confirmed by FISH.

These control experiments were added (Supplementary Fig. 3, 13).

5. I did not fully understand what kind of background subtraction did the authors apply to the FLIM images (Fig 1a, Fig2a, Fig3a).

We thank the reviewer for this important comment. The background subtraction for intensity-based imaging done involved selection of a representative region of the image of background photons i.e. areas where no fluorescence was detected and using this value in standard background subtraction producing final images. For FLIM, we have extracted the lifetime values collected with an IRF standard without subtraction of a background signal and compared the lifetime values for the ROIs across conditions to ensure that we did not introduce artifacts. We elaborated on the data processing (Results and Discussion paragraph 3, 4, Fig. 2) and have added a comment about background signal to the text (Methods, Fluorescence Lifetime Imaging Microscopy (FLIM)).

However, I would like to see even the free dye background (similar to Fig4b), rather than just plain black color inside the cells.

We have changed the image processing and the free dye signal (without RNA, outside of stress granules) is now included in all images (Fig. 2, 5, 6).

They should use an appropriate LUT for display purposes. The selected regions in the images (Fig 1a, Fig2a, Fig3a) do not even have different shades. Why? It is hard to believe that pixel-by-pixel lifetimes are that homogeneous. The authors should always display pixel-by-pixel lifetimes inside the whole cell, not only in the aggregates.

We thank the reviewer for this important note. We added a figure with an overview of common image processing workflows for FLIM (Fig. 2, Results and Discussion paragraph 3). For cases when we extracted one lifetime number for comparisons across conditions in dot plots, i.e. Fig 3, the average lifetime value is preferred as a readout. However, we agree that the pixel-by-pixel representation is ideal for image presentation. We changed the image display accordingly.

6. Since the authors used only two variants of Riboglow (A and D) for imaging two distinct RNAs, it may not be appropriate to write “multiplexed visualization of RNA” in the title and the manuscript. If they create a library of Riboglow mutants with different lifetimes, then the current title would be more appropriate.

We changed the title of the manuscript and text to avoid “multiplexing”.

7. Figure 3: Why is the fluorescence lifetime inside the blue circles even lower than the lifetime of the free probe?

We thank the reviewer for this important observation. Qualitatively, we also observed a slight decrease in lifetime in SGs where NORAD RNA was excluded from SGs compared to the average lifetime of cells with just the probe.

We performed an analysis to systematically compare fluorescence lifetimes in the cytosol, stress granules, and untransfected cells (Supplementary Fig. 17) and did not observe a statistically significant effect.

Reviewer #3 (Remarks to the Author):

The paper titled 'Multiplexed visualization of RNA dynamics in live mammalian cells by fluorescence lifetime imaging microscopy (FLIM) by Sarfraz et al demonstrates the capacity of FLIM to quantitatively image a novel RNA biosensor called Riboglow. This resource has the potential to be very useful for quantification of RNA localisation with respect to sub-cellular compartments, but additional experimental detail and controls experiments would strengthen this application.

Major Comments

1. Riboglow is dim in the RNA unbound state and fluorescent in the RNA bound state and the hypothesis is that this probe's lifetime will increase upon RNA binding. In Fig 1 it is demonstrated that for a given Riboglow construct the average fluorescence lifetime is homogeneous throughout the cell. Is the expectation that the fraction of RNA bound / unbound in each pixel is equal throughout the cell?

We thank the reviewer for this important comment. First, we performed FISH control experiments (Supplementary Fig. 3, 7, 13) to evaluate localization of the tagged mRNA. We also elaborate FLIM data processing (Fig. 2a, Results and Discussion paragraph 3). The homogeneity illustrated in this paper was produced by conducting a multiexponential reconvolution fitting method and extracting a final representative fluorescence lifetime. We use this analysis as the basis to find one number for dot plots (for example, Fig. 3). We acknowledge that using this average number for visualization causes confusion. Therefore, we use pixel-by-pixel fits as a valuable approach when visualizing lifetime as a readout for RNA localization in cells (for example, Fig. 2a, 5, 6). Pixel-by-pixel visualization is more intuitive when looking at specific localizations. The text and figures were updated accordingly.

2. In Fig. 2 it would be useful to see an intensity image of the HaloTag-G33BP1 and NucBlue signal before arsenite stress (like in Fig S5) alongside an unmasked intensity and lifetime map of ACTB-Ribo(4D)-590, and then the stressed example with inclusion of the un-masked intensity and lifetime image of ACTB-Ribo(4D)-590, to assess if intracellular variation exists prior to stress, and assess the impact of these additional fluorescent labels on the lifetime map of ACTB-Ribo(4D)-590. Especially given the potential for FRET as explained in the next comment.

We have made the suggested and important change in figure representation for the stressed condition (now Fig. 5, 6, Supplementary Fig. 14).

1. In Fig. 2-4 stress granules are labelled with Halo-G33BP1-JF646, which has an excitation

spectrum that overlaps with the emission spectrum of ACTB-Ribo(4D)-590 or 1/8-NORAD-Ribo(4D)-590. There is therefore a potential for a FRET interaction between these constructs that would lead to a quenching of the ACTB-Ribo(4D)-590 or 1/8-NORAD-Ribo(4D)-590 lifetime at stress granules. Was this potential artefact eliminated with respect to Fig 4 for example, where a shortened lifetime is indeed observed at stress granules? If not it should be demonstrated that FRET does not impact the measurement.

This is a very important comment. We added a control experiment to eliminate the potential for FRET between JF646 and ATTO 590 where we produced G3BP1 as a fusion with GFP. Cells expressing GFP-G3BP1 as opposed to Halo-G3BP1-JF646 were evaluated to test if the JF646 dye was producing possible artifacts in stress granule evaluation. We found no difference in the lifetimes in stress granules in cells expressing GFP-G3BP1 and cells expressing Halo-G3BP1-JF646 and thus eliminating the concern of artefacts being produced by Halo-G3BP1-JF646 (Supplementary Fig. 16, Results and Discussion paragraph 12).

Minor Comments

1. I found the level of detail provided with respect to the methodology to be limited and sometimes difficult to understand. In many cases this detail is embedded in the supplementary information, but it would be good to provide these details in some capacity in the main manuscript to enable better understanding. For example, from the outset of the results, it would be good to state that the probe fluorophore used is Atto590 and clarify what is the purpose of the mCh protein control experiment for FLIM data processing, given that the IRF was calibrated with Rho B.

We thank the reviewer for this important comment. We made changes to the text accordingly by moving certain figures from the supplement to the main manuscript, as well as adding new figures (Fig. 2, 4). Text has been edited to show ATTO 590 fluorophore was used from the outset of the manuscript (Results and Discussion paragraph 1). mCherry protein control was further explained and highlighted (Results and Discussion paragraph 3). We added more details about data analysis (i.e., Methods section "Fluorescence Lifetime Imaging Microscopy (FLIM)").

4. I found the results presented with respect to Fig 1 confusing since sample i versus iv are described first then sample iii and then finally ii. Potentially it would make more sense to either explain what all four samples are being analysed from the outset (i, ii, iii, iv), then explain in order of presentation, or rearrange figure one such that the sample order is i, iv, iii, ii. Note, the non-significant result described with respect to ii vs iv in Fig. 1c is not labelled in the plot.

We added the labels, and we re-arranged this figure to streamline the data presentation by splitting the figure producing a clearer flow (Fig. 1, 2 and 3).

REVIEWER COMMENTS

Reviewer #1 (Remarks to the Author):

Overall, the authors have addressed the comments well. They have attempted more in vitro and in vivo experiments to improve their lifetime statistics. They have included the data obtained from different cell types, which is now statistically and quantitatively convincing. The images were also improved, and they became much clearer. They also discuss potential problems from artefacts of overexpression conditions, which is helpful for future studies in the field. This is the first report showing two individual RNA imaging by orthogonal RNA aptamer tags, and I recommend it for publication.

Reviewer #2 (Remarks to the Author):

The authors have added a considerable amount of new data and analysis that have noticeably improved and enriched the manuscript. However, there are still major concerns that need to be addressed.

1. Supplementary Table 6 is missing. It appears that the authors used the same FISH probes to visualize both endogenous RNAs (ACTB and NORAD) in untransfected cells and Riboglow-tagged ones in transiently transfected cells (Supplementary Figures 3, 7 and 13). To demonstrate the localization of Riboglow-tagged RNAs and confirm that Riboglow does not affect the localization of the RNA of interest, the authors should use FISH probes targeting the ACTB sequence in untransfected cells and FISH probes targeting the Riboglow sequence in transiently transfected cells. The same is true for NORAD constructs.

2. The new data suggest that Ribo-FLIM does not correctly report the subcellular localization of RNAs. As shown in Figure 1, the average lifetime of free Cbl-5xPEG-ATTO 590 in the nuclei of untransfected cells is approximately 1.4 ns (shown as yellow-green false-color). The FISH images in Supplementary Figure 3 clearly show that both ACTB and NORAD are cytosolic and the most of the nucleus is free of ACTB and NORAD. Therefore, I would expect to see predominantly yellow-green nuclei in the images in Figure 5, 6 and 7. Yet, I don't see any difference between the nucleus and cytosol. Moreover, when ACTB transcripts accumulate in the SGs after arsenite treatment, the contrast between the SG and cytosol is very low. This is probably due to the overexpression of ACTB, because the FLIM-based image is independent of the transcript concentration. In this case, the fluorescence intensity-based images can easily detect the accumulation of RNAs in stress granules and are probably superior to Ribo-FLIM?

3. I strongly recommend that the authors also image other RNAs and verify that Ribo-FLIM can accurately report their localizations.

Minor:

4. Figure 1 and 2 should be combined because Figure 1 does not show anything new. The first figure should ideally present the novelty and the method used in this study.

5. I still find the average lifetime images in Figure 2b confusing and recommend deleting them. Average lifetimes (1.40 and 1.78 ns) can be shown on (or under) the component lifetime image.

6. The y-axis of the graphs in Figure 2c (and throughout the manuscript) should be “fluorescence intensity (au)” and “average lifetime (ns)”

Reviewer #3 (Remarks to the Author):

All of my comments have been addressed the manuscript reads very well.

Point by point response to reviewers, Sarfraz et al, 2023:

Reviewer #1 (Remarks to the Author):

Overall, the authors have addressed the comments well. They have attempted more in vitro and in vivo experiments to improve their lifetime statistics. They have included the data obtained from different cell types, which is now statistically and quantitatively convincing. The images were also improved, and they became much clearer. They also discuss potential problems from artefacts of overexpression conditions, which is helpful for future studies in the field. This is the first report showing two individual RNA imaging by orthogonal RNA aptamer tags, and I recommend it for publication.

We thank reviewer 1 for the positive feedback.

Reviewer #2 (Remarks to the Author):

The authors have added a considerable amount of new data and analysis that have noticeably improved and enriched the manuscript. However, there are still major concerns that need to be addressed.

We thank reviewer 2 for the substantive feedback that undoubtedly improved our manuscript. Detailed responses are included below.

1. Supplementary Table 6 is missing.

We apologize for this oversight, table 6 is now included.

It appears that the authors used the same FISH probes to visualize both endogenous RNAs (ACTB and NORAD) in untransfected cells and Riboglow-tagged ones in transiently transfected cells (Supplementary Figures 3, 7 and 13). To demonstrate the localization of Riboglow-tagged RNAs and confirm that Riboglow does not affect the localization of the RNA of interest, the authors should use FISH probes targeting the ACTB sequence in untransfected cells and FISH probes targeting the Riboglow sequence in transiently transfected cells. The same is true for NORAD constructs.

We thank the reviewer for recommending this important control experiment. To complete thorough validation of the localization of our RNA reporters, we added FISH control experiments for all suggested conditions and added text in the results section to reflect this. Panels with new FISH results are now included in Supplementary Fig. 3. Figure captions were updated in Supplementary Fig. 7 and 13 accordingly.

In summary, we find that the Riboglow tag does not affect localization of the RNA. This confirms that our model system is suitable for assessing Riboglow-FLIM as a new RNA imaging platform.

We would also like to point out that we considered the possibility that during RNA degradation the Riboglow tag would remain and localize to stress granules as an artifact in our stress granules assay. Reassuringly, we observed no unexpected fluorescence lifetime signal in stress granules for the 1/8NORAD-Ribo(4A)-590 construct (Supplementary Fig. 17). This particular truncated NORAD variant is excluded from stress granules, as we and others (Matheny et al, 2021, RNA). If the Riboglow tag would remain as a degradation product and produce a nonspecific fluorescence lifetime signal in stress granules, we would observe this for these samples.

2. The new data suggest that Ribo-FLIM does not correctly report the subcellular localization of RNAs. As shown in Figure 1, the average lifetime of free Cbl-5xPEG-ATTO 590 in the nuclei of untransfected cells is approximately 1.4 ns (shown as yellow-green false-color). The FISH images in Supplementary Figure 3 clearly show that both ACTB and NORAD are cytosolic and the most of the nucleus is free of ACTB and NORAD. Therefore, I would expect to see predominantly yellow-green nuclei in the images in Figure 5, 6 and 7. Yet, I don't see any difference between the nucleus and cytosol.

The reviewer is correct that the live images we presented appear to indicate an unexpected behavior. Indeed, our FISH control dataset is consistent with images of mRNA localization in the literature and the FLIM images appear to report a different localization.

First, we are grateful for the suggestion of the thorough FISH control experiments (explained above), as these give us confidence that the transiently expressed ACTB mRNA and NORAD constructs (tagged with Riboglow) show localization patterns one would expect from observations in the literature.

Second, we now realize that we did not carefully control the z-axis in collecting live cell fluorescence images. We realize that some images were collected when the z-plane was not perfectly adjusted to go through the nucleus. This contributed to some fluorescence signal that suggest reporting on nuclear signal, when we collected images in a cytosolic plane slightly above or below the nucleus. The fixation process eliminates this z-height complication, which is why this was not observed for the FISH images. We addressed this concern as follows. For the dataset that compares the cytosol and nucleus fluorescence lifetime, we revisited which cells we displayed to avoid this misrepresentation (Supplementary Fig. 8). Representative FLIM live cells in this figure now illustrate the point that the lifetime signal in the nucleus and cytosol is different and in line with FISH images, as expected by reviewer 2 (i.e., Supplementary Fig. 8b). In this context, it is important to note that we specifically chose model RNAs with cytosolic localization patterns (and not nuclear) throughout this work.

Third, we carefully evaluated fluorescence lifetime numbers of our Riboglow probe alone (in the absence of RNA, i.e. untransfected cells). In these untransfected control samples, we observed a slightly higher lifetime value in the nucleus compared with the cytosol (Supplementary Fig. 8). Importantly, this slightly elevated lifetime value was observed in untransfected cells and those that produce ACTB mRNA, as the reviewer pointed out. In both cases, no Riboglow RNA is present in the nucleus and the lifetime values are indeed indistinguishable. This control analysis confirms to us that slightly elevated nuclear lifetime values do not originate from RNA mislocalization. This effect slightly complicates the differentiation of nuclear vs. cytosolic FLIM data and requires additional controls to adjust reported FLIM values when investigating nuclear

RNA with Riboglow-FLIM. For this study, we chose to use only cytosolic model RNAs to avoid these complications. A thorough investigation of nuclear FLIM effects may be the focus of a follow-up study.

More generally, a slight increase in fluorescence lifetime when a fluorescence sensor is placed in the nucleus vs. the cytosol is a well-established observation in the FLIM field, as observed in the following references:

- Okabe, K., Sakaguchi, R., Shi, B. et al. Intracellular thermometry with fluorescent sensors for thermal biology. *Pflugers Arch - Eur J Physiol* 470, 717–731 (2018), <https://doi.org/10.1007/s00424-018-2113-4>
- Okabe, K., Inada, N., Gota, C. et al. Intracellular temperature mapping with a fluorescent polymeric thermometer and fluorescence lifetime imaging microscopy. *Nat. Commun.* 3, 705 (2012), <https://doi.org/10.1038/ncomms1714>
- Pham, N, N, T, Kang, S. G., Son, Y.-A. et al. Tolerance of Iron Phthalocyanine Functionalized with Graphene Quantum Dots: A Density Functional Theory Approach, *J. Phys. Chem. C* 2019, 123, 45, 27483–27491, <https://doi.org/10.1021/acs.jpcc.9b06750>
- Huang, M., Liang, X., Zhang, Z. et al. Carbon Dots for Intracellular pH Sensing with Fluorescence Lifetime Imaging, *Microscopy. Nanomaterials.* 10. 604 (2020), <https://doi.org/10.3390/nano10040604>
- Dai, X., Yue, Z., Eccleston, M. E., et al, Fluorescence intensity and lifetime imaging of free and micellar-encapsulated doxorubicin in living cells, *Nanomedicine: Nanotechnology, Biology and Medicine*, 4, 1, 49-56, (2008), <https://doi.org/10.1016/j.nano.2007.12.002>.
- Mueller-Harvey, I., Feucht, W., Polster, J., et al. Two-photon excitation with pico-second fluorescence lifetime imaging to detect nuclear association of flavanols, *Analytica Chimica Acta*, 719, 68-75 (2012), <https://doi.org/10.1016/j.aca.2011.12.068>.

This is likely caused by nuclear compaction or other environmental differences between cellular compartments like temperature or volume effects. Importantly, these small nucleus/cytosol differences are much smaller than probe +/- RNA lifetimes we observed.

Moreover, when ACTB transcripts accumulate in the SGs after arsenite treatment, the contrast between the SG and cytosol is very low. This is probably due to the overexpression of ACTB, because the FLIM-based image is independent of the transcript concentration. In this case, the fluorescence intensity-based images can easily detect the accumulation of RNAs in stress granules and are probably superior to Ribo-FLIM?

This is an interesting suggestion that we appreciate. In fact, our FLIM datasets already include intensity values. We now report the fluorescence lifetime and intensity values for the suggested conditions side by side (Supplementary Fig. 14). Together, we find that the contrast for cytosol / nucleus is more robust when analyzed by FLIM compared with intensity. We thank the reviewer

for pointing out to remain agnostic regarding the use of intensity vs. lifetime imaging. Long term, we envision that Riboglow-FLIM serves as one additional RNA imaging tool in the field that may be useful for some applications, but we do not think it should replace intensity-based imaging. We suggest that detailed side-by-side comparisons are a key foundation to guide users in the future.

3. I strongly recommend that the authors also image other RNAs and verify that Ribo-FLIM can accurately report their localizations.

We understand that the reviewer is concerned that Riboglow-FLIM may not accurately report on RNA subcellular localizations live. We believe that we added thorough control FISH experiments (outlined above) to address this concern. Furthermore, we now include substantial additional analyses of cytosolic vs. nuclear RNA localizations for Riboglow-FLIM that addresses this concern (point 2, above). We are not convinced that tagging additional RNAs with Riboglow will add new insights to this study and address the concern about detailed subcellular localization reporters. Therefore, we believe that tagging additional RNAs is outside of the scope of this work.

Minor:

4. Figure 1 and 2 should be combined because Figure 1 does not show anything new. The first figure should ideally present the novelty and the method used in this study.

This is an interesting suggestion. We prefer to leave figures 1 and 2 separately, since figure 2 is already quite substantial. We want to point out that the FLIM application is included in figure 1 that illustrates the novelty and method used in this study.

5. I still find the average lifetime images in Figure 2b confusing and recommend deleting them. Average lifetimes (1.40 and 1.78 ns) can be shown on (or under) the component lifetime image.

We agree with the reviewer about the possible confusion. We find it useful to illustrate the two different lifetime analyses not just in cartoon form (2a) but also in the image (2b).

6. The y-axis of the graphs in Figure 2c (and throughout the manuscript) should be “fluorescence intensity (au)” and “average lifetime (ns)”

We suspect that the reviewer is pointing out the ambiguity of FLIM data analysis. We have expanded all figure legends to make clear that we mean “average fluorescence lifetime” in dot plots, as defined in Fig. 2. We clarified the language in the figure legend for Fig. 2 and all figure legends throughout this study to ensure consistency. We also added text in the main manuscript to clarify this point.

Reviewer #3 (Remarks to the Author):

All of my comments have been addressed the manuscript reads very well.

We thank reviewer 3 for this positive feedback.